

**Distribution, chemical and molecular composition of high and low-**
**molecular-weight humic-like substances in ambient aerosols**
Xingjun Fan[a,b,]*, Ao Cheng[a], Xufang Yu[a], Tao Cao[b,c], Dan Chen[a], Wenchao Ji[a],
Yongbing Cai[a], Fande Meng[a], Jianzhong Song[b,]**, Ping'an Peng[b]
[a] College of Resource and Environment, Anhui Science and Technology University,
Fengyang 233100, P. R. China
[b] State Key Laboratory of Organic Geochemistry, Guangzhou Institute of Geochemistry,
Chinese Academy of Sciences, Guangzhou 510640, P. R. China
[c] University of Chinese Academy of Sciences, Beijing, 100049, PR China
* Corresponding authors
E-mail addresses: fanxj@ahstu.edu.cn (Xingjun Fan), songjzh@gig.ac.cn (Jianzhong
Song)





**Abstract**
Humic-like Substances (HULIS) encompass a continuum of molecular weight
(MW) ranges, yet our understanding of how HULIS characteristics vary with MW is
still limited and not well-established. In this study, a combination of ultrafiltration and
solid-phase extraction protocols was employed to fractionate the high MW (HMW, >1
kDa) and low MW (LMW, < 1kDa) HULIS fractions from ambient aerosols collected
during summer and winter at a rural site. Subsequently, comprehensive characterization
by using total organic carbon, high-performance size exclusion chromatography
(HPSEC), UV-vis and fluorescence spectroscopy, Fourier-transform infrared
spectroscopy (FTIR), negative electrospray ionization high resolution mass
spectrometry (ESI- HRMS) were conducted. The results revealed that HMW HULIS
were dominated by larger-sized chromophores, substantially constituting a higher
fraction of total organic carbon and UV absorption at 254 nm than LMW HULIS. While
both HMW and LMW HULIS shared similar fluorophore types and functional groups,
the former exhibited higher levels of humification and a greater presence of polar
functional groups (e.g., -COOH, >C=O). HRMS analysis further unveiled that
molecular formulas within HMW HULIS generally featured smaller sizes but higher
degrees of unsaturation and aromaticity compared to those within LMW HULIS
fractions. This observation suggests the possibility of small molecules assembling to
form the HMW HULIS through intermolecular weak forces. Moreover, HMW HULIS
contained a higher proportion of CHON but fewer CHO compounds than LMW HULIS.
In both HMW and LMW HULIS, the unique molecular formulas were primarily



characterized by lignin-like species, yet the former displayed a prevalence of N-
enriched and highly aromatic species. Additionally, HMW HULIS contained more
unique lipids-like compounds, while LMW HULIS exhibited a distinct presence of
tannin-like compounds. These findings provide valuable insights into the distribution,
optical properties, and molecular-level characteristics of HULIS in atmospheric
aerosols, thereby advancing our understanding of their sources, composition, and
environmental implications.

**Keywords:** Humic-Like Substances, molecular weight fractionation, optical properties,
high-performance size exclusion chromatography, negative electrospray ionization-
high resolution mass spectrometry



**1. Introduction**

HUmic-Like Substances (HULIS) are complex and heterogeneous mixtures of water-soluble organic matters (WSOM) that are of great importance in the atmospheric environment. They usually share similar physicochemical properties (e.g., acidity, absorption, fluorescence, functional groups) with naturally occurring humic substances (Graber and Rudich, 2006; Zheng et al., 2013) and are prevalent in fog, clouds, rainwater and ambient aerosols (Birdwell and Valsaraj, 2010; Fan et al., 2016a; Santos et al., 2012). With substantial hygroscopic and surface-active properties, HULIS enhance the hygroscopic growth of particles, thereby contributing to the formation of the cloud condensation nuclei and ice nuclei (Chen et al., 2021a; Dinar et al., 2007). Moreover, acting as an important component of brown carbon (BrC), HULIS effectively absorb near-ultraviolet and visible light, thus influencing the global radiative balance and atmospheric chemistry processes (Bao et al., 2022; Zhang et al., 2020). Furthermore, HULIS have the potential to catalyze the formation of reactive oxygen species, leading to potential adverse health effects (Ma et al., 2019; Zhang et al., 2022b).

The chemical composition of atmospheric HULIS exhibit significant heterogeneity and typically comprises macromolecular compounds containing aromatic rings with highly conjugated structures, as well as long-chain hydrocarbon with polar groups (e.g., -OH, -COOH, -NO2) (Fan et al., 2013; Huo et al., 2021). To unravel the structural characteristics and properties of HULIS, a range of analytical techniques, including absorption and fluorescence spectroscopy, Fourier transform infrared spectroscopy (FTIR), nuclear magnetic resonance spectroscopy ($^1$H NMR), have been



utilized (Huo et al., 2021; Qin et al., 2022; Zou et al., 2020). These studies have
provided insights into the overall structural characteristics of complex HULIS,
including their abundances, chemical and optical characteristics (Huo et al., 2021;
Mukherjee et al., 2020; Win et al., 2018; Zhang et al., 2022b; Zheng et al., 2013). In
recent years, high-resolution mass spectrometry (HRMS) techniques, such as Fourier
transform ion cyclotron resonance mass spectrometry (FT-ICR-MS) and orbitrap
HRMS, in combination with electrospray ionization (ESI), have emerged as powerful
tools for elucidating the molecular-level characteristics of HULIS (Lin et al., 2012; Sun
et al., 2021; Wang et al., 2019; Zou et al., 2023). By utilizing HRMS, researchers have
gained deeper insights into the complexity and chemical heterogeneity of HULIS at the
molecular level.
Operationally, HULIS are defined as the hydrophobic fraction of water-soluble
organic matter (WSOM) typically extracted through solid-phase extraction (SPE)
protocol (Fan et al., 2012; Zou et al., 2020). Thus, the abundance and characteristics of
HULIS are contingent upon the chemical composition of WSOM. Previous studies have
shown that aerosol WSOM, also known as brown carbon (BrC), are comprised of a
continuum of molecular weight (MW) species, as revealed by high-performance
exclusion chromatography (HPSEC) analysis (Di Lorenzo et al., 2017; Fan et al., 2023;
Wong et al., 2019). These studies have highlighted that BrC typically consist of both
high-MW (HMW) and low-MW (LMW) chromophores in various aerosols. For
example, BrC emitted from fresh biomass burning (BB) are dominated by low MW
chromophores (Di Lorenzo et al., 2017; Wong et al., 2019). However, BrC derived from



aged BB aerosols and ambient aerosols tend to possess more HMW chromophores that
are highly chemically resistant (Di Lorenzo et al., 2017; Fan et al., 2023; Wong et al.,
2019). Further characterizations of different MW BrC can be conducted using an
ultrafiltration (UF) protocol (Fan et al., 2021). This approach enabled researchers to
obtain the distributions of content, chromophores and fluorophores within various MW
BrC fractions. Despite these advancements, the chemical structures and molecular
composition of different MW HULIS fractions remain poorly understood.
Consequently, a combination of UF and SPE protocols for the fractionation and
characterization of MW-separated HULIS is crucial, as it not only provides insights into
MW distributions but also illuminates the chemical heterogeneities of aerosols HULIS.

In this study, a combination of UF-SPE isolation protocol was developed to

fractionate and characterize the MW HULIS fractions. Two distinct sets of ambient
$PM_{2.5}$ samples collected during summer and winter periods were utilized to facilitate a
comparative analysis of MW HULIS. Initially, the WSOM were fractionated into high-
MW (HMW, >1 kDa) and low-LMW (LMW, <1 kDa) species using the UF protocol.
Subsequently, the resulting MW MSOM fractions underwent SPE to obtain different
MW HULIS fractions. The obtained HMW and LMW HULIS fractions were
comprehensively characterized using advanced analytical techniques, including total
organic carbon analysis, UV-vis and fluorescence spectroscopy, HPSEC, and HRMS to
unveil their abundances, absorption and fluorescence properties, and molecular
characteristics. The findings of this study hold great significance in advancing our
understanding of the definition and molecular profiles of HULIS, as well as facilitating





further investigations into their potential impacts on the atmospheric environment.

**2. Materials and methods**
2.1. Atmospheric fine particles sampling
Atmospheric $PM_{2.5}$ were sampled on the rooftop of a building within the campus
of Anhui Science and Technology University (32.21ºN, 118.72ºE), around 20 m above
ground level. Detailed information regarding the sampling site can be found in our
previous studies (Cao et al., 2022; Fan et al., 2021). The $PM_{2.5}$ samples were collected
using a high-volume $PM_{2.5}$ sampler (JCH-1000, Juchuang Ltd., Qingdao) onto
prebaked quartz fiber filters (8 × 10 inches, Whatman). Sampling took place from July
25 to August 12, 2021, during summer, and from December 19, 2021 to January 6, 2022,
during winter. Blank filters were also collected as control samples. All aerosol $PM_{2.5}$
filter samples were stored at -20 ºC in a freezer prior to analysis. The atmospheric
pollutant data ($NO_2$, $SO_2$ and $O_3$) near sampling site during sampling period were
obtained from the website (https://www.aqistudy.cn) and are summarized in Table S1.
Additionally, the fire spots were investigated using data from the website
(https://firms.modaps.eosdis.nasa.gov/map/) and are visualized in Fig. S1.

2.2. Application of UF-SPE for isolating MW HULIS fractions
Punches of summer and winter aerosol $PM_{2.5}$ filter samples were taken and
combined for the extraction of water-soluble organic matter (WSOM), respectively. The
filters were immersed in 300 mL of ultrapure water and subjected to ultrasonication for





30 min. The resulting suspensions were then filtered through a 0.22 μm membrane (Φ
47 mm, Jinteng, China) to obtain bulk WSOM samples. These bulk filtrates were further
subjected to UF and SPE in tandem to obtain different MW HULIS fractions. Please
refer to Fig. S2 for a schematic representation of the fractionation steps.
Before UF, the bulk WSOM were diluted to DOC concentration below 30 mg/L to
minimize the concentration effects and prevent the accumulation of organic matters at
the membrane surface during UF. The detailed UF procedure followed the profile
described in our previous study (Fan et al., 2021). Briefly, each bulk WSOM solution
was passed through a pre-cleaned 1 kDa cut-off membrane in a stirred UF cell (Amicon
8200, Millipore, USA), with a pressure at 0.2 MPa applied by ultrapure $N_2$. The
concentration factor was ~10. The resulting retentate was considered as HMW (>1 kDa)
WSOM, while the permeate solutions represented the LMW (<1 kDa) WSOM. Finally,
each MW fraction was diluted to the initial volume for further treatment and analysis.
Mass balances of WSOM during one-step UF process generally ranged from 92% to
99%, as determined by total organic carbon (TOC) and UV absorption at 254 nm
($UV_{254}$), indicating good performance of UF without substantial loss or organic
contamination.
Subsequently, SPE was applied to isolate the so-called HULIS fractions from the
bulk and each MW fraction of WSOM, following the protocol proposed in our previous
studies (Fan et al., 2013; Zou et al., 2020). Briefly, the acidified aqueous samples were
passed through pre-activated HLB columns (Waters Oasis, 500 mg/6 mL, USA). The
fractions retained on the resins (referred to as HULIS) were eluted with pure methanol

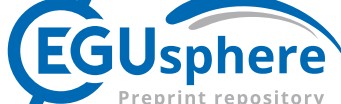

and dried using a gentle stream of pure $N_2$. Finally, the bulk, HMW and LMW HULIS
fractions were obtained. A blank filter control was performed using the same procedure
described above, and the analysis signals of samples were corrected by blank control.

2.3. HPSEC analysis
The apparent MW distributions of MW HULIS fractions were analyzed using a
high-performance liquid chromatography (HPLC) system (LC-20AT, Shimadzu, Japan)
equipped with a refractive index detector (RID-10A, Shimadzu) and a diode array
detector (SPD-M20A, Shimadzu). The wavelength of the diode array detector was set
at 254 nm. Separation was performed using an aqueous gel filtration column (Polysep-
GFC-P 3000, Phenomenex) preceded by a guard column (Polysep-GFC-P,
Phenomenex). The mobile phase consisted of a mixture of water and methanol (90:10
v/v) containing 25 mM ammonium acetate (Di Lorenzo et al., 2017; Wong et al., 2019).
The sample injection volume was 100 μL, and the flow rate was maintained at 1 mL
$min^{-1}$. The HPSEC calibration was performed using a series of polyethylene glycol
(PEG) standards (Kawasaki et al., 2011; Zhang et al., 2022c). The chromatographic
peak areas were integrated to represent the abundances of corresponding MW species.
It should be noted that the MW values estimated here are nominal rather than absolute.
The weight-average MW (Mw), number-average MW (Mn) and polydispersivity
(ρ), were determined using the following equations (Song et al., 2010):

$$Mw = \frac{\sum_{i=1}^{n}(h_i MW_i)}{\sum_{i=1}^{n} h_i} \qquad (1)$$

$$Mn = \frac{\sum_{i=1}^{n} h_i}{\sum_{i=1}^{n}(h_i/MW_i)} \qquad (2)$$

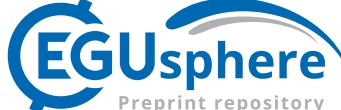

$$\rho = \frac{Mw}{Mn} \qquad\qquad (3)$$

where $h_i$ and $MW_i$ are the absorption intensity of the chromatogram and the MW of
molecules corresponding to the $i$th retention time, respectively.

2.4. Measurements of WSOC content and optical properties
The concentration of water-soluble organic carbon (WSOC) in HMW and LMW
HULIS was measured using a Shimadzu TOC analyzer (TOC-VCPN, Japan) following
the non-purgeable organic carbon protocol.
The UV-vis spectra were recorded using a UV-vis spectrophotometer (UV 2600,
Shimadzu, Japan) over a wavelength range of 200-700 nm with 1 nm increments.
Excitation-emission matrix (EEM) spectra were determined using a fluorescence
spectrophotometer (F4600, Hitachi, Japan). The scanning ranges for excitation (Ex) and
emission (Em) wavelengths were 200-400 and 290-520 nm, respectively, with a
scanning speed was 12,000 nm/min.
To characterize the chemical and optical properties of MW HULIS fractions,
several commonly used spectra parameters were calculated, including the specific UV
absorbance at 254 nm (SUVA$_{254}$), the UV absorbance ratio between 250 and 365 nm
($E_2/E_3$), spectra slope ratios ($S_R$), the absorption Angstrom exponent (AAE), and mass
absorption efficiency (MAE$_{365}$), fluorescence indices (FI), biological index (BIX), and
humification degree (HIX) (Fan et al., 2021; Li et al., 2022; Wu et al., 2021). Further
details can be found in Text S1 of the Supporting Information (SI).



2.5. FTIR spectrometry

The functional groups in HMW and LMW HULIS were characterized using a Nicolet iS50 FTIR spectrometer (Thermal Fisher, USA). Before analysis, the freeze-dried MW HULIS and pure KBr were thoroughly mixed, finely ground, and pressed into pellets under dry conditions. Then, the FTIR spectra of samples were recorded within the wavenumbers ranging from 4000 to 400 $cm^{-1}$, with a resolution of 4 $cm^{-1}$. To ensure accuracy, each spectrum was baseline-corrected using the pure KBr spectrum.

2.6. HRMS analysis and data processing

The MW HULIS fractions were analyzed using a Q-Exactive mass spectrometer (Thermo Scientific, Germany) equipped with a heated electrospray ionization (ESI) source. The system operated in negative ESI mode with a resolution of 140,000 at m/z = 200. The detection mass range was set from 60 to 900 m/z. To ensure accurate mass measurements, mass calibration was carried out using a commercial standard mixture of ESI-L Low Concentration Tuning Mix (G1969-85000, Agilent, USA).

The acquired mass spectra were processed using Xcalibur software (V2.2, Thermo Scientific). The mathematically possible formulas for all ions were calculated with a signal-to-noise ratio (s/n) $\geq$ 5 using a mass tolerance of 5 ppm. The assigned molecular formulas followed specific constraints, with limitations on the following elements: $C \leq 50$, $H \leq 100$, $O \leq 20$, $N \leq 3$, $S \leq 2$. Additionally, the elemental ratios of H/C, O/C, N/C, and S/C were constrained to the ranges of 0.3–3.0, 0–3.0, 0–0.5, and 0–2.0, respectively. The double-bond equivalents (DBE) and modified aromaticity index



($AI_{mod}$) values of the assigned neutral assigned formula ($C_cH_hO_oN_nS_s$) were calculated
using the equations (4-5) (He et al., 2023; Song et al., 2022):

$$DBE = 1 + \frac{1}{2}(2c - h + n) \tag{4}$$

$$AI_{mod} = \frac{1 + c - 0.5o - 0.5n - 0.5h}{c - 0.5o - n} \tag{5}$$

The intensity-weighted molecular parameters ($X_w$) of MW, H/C, O/C, DBE, and
AI values were calculated according to the equation (6) (He et al., 2023; Zhang et al.,
2021; Zou et al., 2023):

$$X_W = \frac{\sum(I_i \cdot X_i)}{\sum I_i} \tag{6}$$

where $X$ represents the aforementioned parameters, and $I_i$ denote the intensity for each
assigned formula $i$.

**3. Results and discussion**
3.1. Size and distribution of MW HULIS fractions
3.1.1. Molecular size of HMW and LMW HULIS
Fig. 1 shows the HPSEC chromatograms of MW HULIS. Both HMW and LMW
HULIS exhibit MW continuum distributions ranging from 100 to 20,000 Da, which is
consistent with the reported distributions of BrC in BB-derived and various ambient
aerosol in previous studies (Di Lorenzo et al., 2017; Fan et al., 2023; Wong et al., 2017).
However, the chromatographic patterns for HMW HULIS clearly differ from those
observed for LMW HULIS in both aerosol samples. As seen in Fig. 1, HMW HULIS
display an additional and stronger absorption peak at around 4000 Da (peak *iii*), along
with a more pronounced peak at 2200 Da (peak *ii*) and a similar magnitude peak at 570

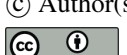

Da (peak *i*) compared to LMW HULIS. This suggests that HMW HULIS contain the
majority of larger molecular size chromophores within the bulk WSOM.

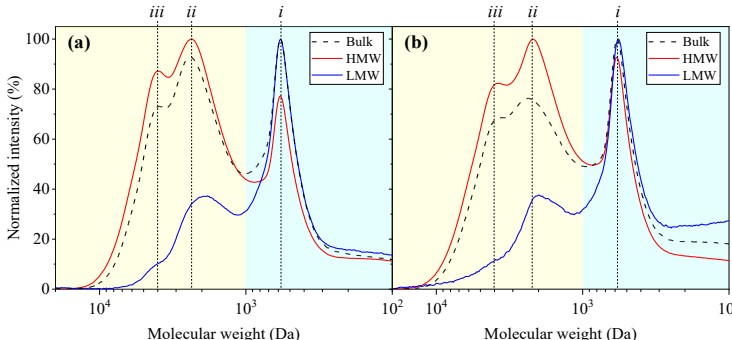


**Fig. 1.** Average HPSEC chromatograms of bulk, HMW and LMW HULIS fractions in
(a) summer and (b) winter aerosol, respectively. The yellow and cyan shadows represent
MW size regions of >1 kDa and <1 kDa, respectively.

Moreover, the molecular size of MW HULIS can be further reflected by the

differences in Mw and Mn. As listed in Table 1, the average Mw and Mn of HMW
HULIS are 2233-2315 and 654-707 Da, respectively, greatly lager than that of LMW
HULIS (989-1071 and 293-394 Da, respectively). These differences indicate that the
sources and formation processes of HMW HULIS may differ from LMW HULIS. Many
previous studies have demonstrated that complex atmospheric aging processes
significantly enhance the formation of large molecular size chromophores, while
concurrently leading to the bleaching of small size ones (Di Lorenzo et al., 2018; Di
Lorenzo et al., 2017; Wong et al., 2017; Wong et al., 2019). Therefore, the higher
proportions of large-size chromophores and resulting larger apparent molecular size of
HMW HULIS may indicate their possible secondary formation nature.




**Table 1.** The summary of typical quantity and quality parameters of each MW HULIS

fraction from BB and ambient aerosols.

| | | Summer | | | Winter | | |
|---|---|---|---|---|---|---|---|
| | | Bulk | HMW | LMW | Bulk | HMW | LMW |
| HPSEC-derived | $M_w$ | 1975±13 | 2315±38 | 1071±24 | 1918±56 | 2233±42 | 989±67 |
| parameters | $M_n$ | 591±53 | 707±48 | 394±13 | 525±57 | 654±17 | 293±32 |
| | ρ | 3.4±0.3 | 3.3±0.2 | 2.7±0.2 | 3.7±0.3 | 3.4±0.2 | 3.4±0.2 |
| HULIS/WSOM | TOC | 65±1 | 68±1 | 51±2 | 63±2 | 67±2 | 41±1 |
| (%)[a] | $UV_{254}$ | 66±5 | 65±2 | 55±4 | 67±1 | 65±1 | 61±2 |
| Optical | $E_2/E_3$ | 12.02±0.54 | 11.72±0.31 | 14.98±0.98 | 6.30±0.24 | 6.54±0.16 | 7.24±0.43 |
| parameters | $MAE_{365}$ | 0.21±0.02 | 0.23±0.01 | 0.20±0.01 | 1.04±0.02 | 1.06±0.01 | 0.88±0.00 |
| | AAE | 7.11±0.32 | 7.59±0.00 | 8.25±0.23 | 6.66±0.06 | 6.25±0.06 | 7.28±0.03 |
| | FI | 2.00±0.04 | 1.99±0.03 | 2.04±0.05 | 2.06±0.01 | 1.97±0.03 | 2.25±0.02 |
| | BIX | 0.95±0.01 | 0.86±0.07 | 1.02±0.01 | 0.96±0.01 | 0.81±0.01 | 1.07±0.02 |
| | HIX | 2.42±0.06 | 2.43±0.04 | 2.40±0.05 | 3.13±0.25 | 5.64±0.34 | 1.94±0.16 |

[a] The ratios of contents of SPE-isolated HULIS fractions to that of corresponding

WSOM fractions determined by TOC and/or absorbance at 254 nm ($UV_{254}$).

3.1.2. Relative abundances of HMW and LMW HULIS

The contribution of MW-HULIS fractions to their corresponding MW-WSOM

fractions, quantified in terms of TOC and UV absorption at 254 nm for both summer

and winter aerosols are summarized in Table 1. In general, the ratios of HULIS/WSOM

of HMW fractions (in terms of TOC and UV254) (65-68%) were higher than the ratios

(41-61%) observed for LMW fractions. This finding suggests that the higher presence

of hydrophobic and conjugated aromatic structures in HMW WSOM, but more

hydrophilic OC and non-aromatic species (e.g., aliphatic dicarboxylic acid) in the

LMW WSOM (Fan et al., 2012; Zou et al., 2020).

Fig. 2 illustrates the distribution of distinct MW fractions within reconstructed



WSOM, wherein "non-HULIS" refers to the content differences between the MW
WSOM and its HULIS fractions. The HMW HULIS fraction contributed 39-41% of
TOC and 40-47% of UV254 to the bulk WSOM. In contrast, the LWM HULIS fraction
only make up a smaller proportion, accounting for 16-20% of TOC and 17-21% of
UV254 within the bulk WSOM. Specifically, the ratios between HMW HULIS and
LMW HULIS (H/L) ranged from 1.88 to 2.75 for both summer and winter aerosols in
terms of either TOC or UV254. These findings emphasize that HMW HULIS
significantly dominate the bulk aerosol HULIS fractions. Notably, the H/L ratio for
winter aerosols was higher than that for summer aerosols, suggesting that larger-sized
HULIS contributed more to the bulk HULIS fractions in winter aerosols.

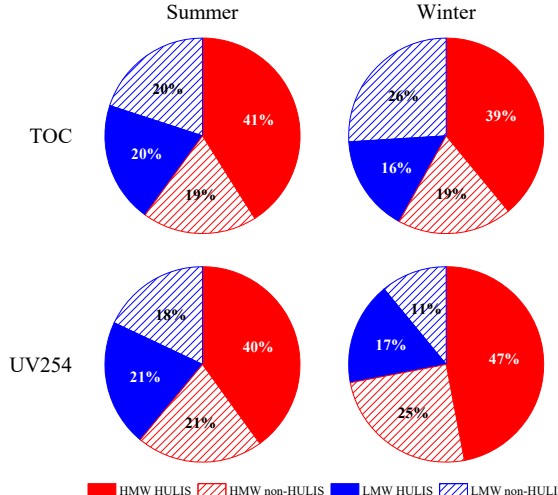

**Fig. 2.** Relative proportions of different MW fractions in summer and winter aerosols
determined by TOC and UV254.

The non-HULIS fractions are also important constituents within aerosol WSOM,

but exhibit some differences between HMW and LMW fractions. The contributions of





HMW non-HULIS to bulk WSOM were ~19% as determined by TOC and 21-25%
measured by UV254. In case of LMW non-HULIS, the contributions were higher in
terms of TOC (20-26%) but lower in terms of UV254 (11-18%). These results indicate
that the LMW WSOM contain a larger proportion of hydrophilic organic species with
weak or no light absorption.

3.2. Optical characteristics of MW HULIS fractions
3.2.1. Light absorption characteristics

The absorption spectra of MW HULIS fractions in ambient aerosols are shown in

Fig. S3. These spectra exhibit a featureless shape with a general decrease in absorbance
as the wavelength increases, which is a typical characteristic of HULIS found in
rainwater, biomass burning (BB), and ambient aerosols (Huo et al., 2021; Santos et al.,
2009; Zhang et al., 2022b). The E2/E3 ratio, commonly used as an indicator of the
chemical characteristics of organic species, is inversely correlated with higher
aromaticity and larger molecular weight (Fan et al., 2021; Li et al., 2022; Sun et al.,
2021). As listed in Table 1, the $E_2/E_3$ of HMW HULIS fractions generally were lower
than that of LMW HULIS in both ambient aerosols. This is consistent with the
expectation that larger-sized HULIS generally possess more polyconjugated and
polymeric structures (Fan et al., 2021; Zhang et al., 2022c), leading to greater
aromaticity and larger molecular size.

$MAE_{365}$ and AAE are commonly used to characterize the light absorption capacity

and the spectral dependence of light absorption by aerosol chromophores, respectively





(Bao et al., 2022; Fan et al., 2016b; Kumar et al., 2017; Yuan et al., 2021; Zou et al.,
2020). As listed in Table 1, the average $MAE_{365}$ values of HMW HULIS are 0.23 and
1.06 $m^2$ $g^{-1}$ in summer and winter aerosol, respectively. These values are higher than
the corresponding values of 0.20 and 0.88 $m^2$ $g^{-1}$, respectively, for LMW HULIS. In
addition, HMW HULIS presented lower AAE values, being 7.59 and 6.25 in summer
and winter aerosol, respectively, than the corresponding values of 8.25 and 7.28,
respectively, for LMW HULIS (Table 1). As shown in Fig. S4, the LMW HULIS exhibit
lower $MAE_{365}$ and higher AAE values, falling within the left-upper range of the values
previously reported for various ambient aerosol-derived and BB-derived HULIS (Bao
et al., 2022; Fan et al., 2018; Fan et al., 2016b; Hong et al., 2022; Huo et al., 2018; Liu
et al., 2018; Ma et al., 2019; Sun et al., 2021; Tang et al., 2020; Wu et al., 2018; Wu et
al., 2020; Yuan et al., 2021; Zhang et al., 2022a). It has been widely reported that
pronounced photooxidation and photobleaching processes of BrC can lead to a
reduction in their absorption capacity (Wu et al., 2018; Wu et al., 2020; Zhang et al.,
2022a), but an enhancement of their spectra dependence on wavelength (Chen et al.,
2021b; Sun et al., 2021). Therefore, it can be speculated that LMW HULIS are more
susceptible to enrich the by-products resulting from the degradation and oxidation of
BrC during processes like photooxidation and photobleaching.

3.2.2. Fluorescence characteristics
The EEM contours of MW HULIS fractions from both summer and winter
aerosols are presented in Fig. S5. These HULIS fractions from both seasons exhibit



similar EEM spectra features, with a predominance of humic-like fluorophores (Ex/Em
= 210-235/395-410 nm). This observation suggest that humic-like fluorophores are
fundamental constituents of both HMW and LMW HULIS, which are consistent with
previous findings for aerosols MW WSOM (Fan et al., 2021) and bulk HULIS in BB-
derived and ambient aerosols (Fan et al., 2020; Qin et al., 2018). In this study, the
fluorescence regional integration (FRI) method was applied to characterize the
fluorescent composition of MW HULIS. Using FRI, EEM spectra were divided five
fluorescence regions (labeled as I to V) (Fig. S5), which were successively assigned to
simple aromatic proteins (I and II), fulvic acid-like (III), soluble microbial byproduct-
like (IV), and humic acid-like (V) substances, respectively, as established in previous
studies (Chen et al., 2003; Qin et al., 2018; Wang et al., 2021b). As shown in Fig. S6,
the large-size aromatic proteins (II) and fulvic acid-like substances (III) dominated the
fluorophores within MW HULIS in both summer and winter aerosols, comprising
approximately 62-64% of the total fluorescence intensity. This finding is consistent
with previous reports on bulk HULIS in summer and winter aerosols from industrial
and urban cities (Qin et al., 2018; Wang et al., 2021b). In comparison, the HMW HULIS
in both summer and winter aerosols generally exhibited a higher proportion of humic
acid-like substances (V), while having a lower abundance of small-size aromatic
proteins I compared to LMW HULIS. These differences are particularly pronounced in
winter aerosols, with the humic acid-like substances accounting for 23% in HMW
HULIS compared to 13% in LMW HULIS, and small-size aromatic proteins I
comprising 9% in HMW HULIS compared to 17% in LMW HULIS (Fig. S6).





Furthermore, the higher HIX values of HMW HULIS (5.64) in comparison to LMW
HULIS (1.94) further support these differences (Table 1). The pronounced BB
emissions and potential $NO_2$-related oxidation of OA, as evidenced by the presence of
more hotspots (Fig. S1) and higher concentration of $NO_2$ (Table S1), are likely driving
these marked distinctions between HMW and LMW HULIS in winter aerosols. In
general, these findings imply that the HMW HULIS have a stronger level of
humification and oxidation, while the LMW HULIS appear to be of a simpler nature
and are more likely associated with fresh emissions (e.g., BB).

3.2.3 Functional groups of MW HULIS
Fig. 3 depicts the FTIR spectra of HMW and LMW HULIS in both summer and
winter aerosols. In general, both HMW and LMW HULIS present similar absorption
peaks, including pronounced peaks at 3434 cm$^{-1}$ (O-H stretching of phenols and
carboxylic acids), 1721 cm$^{-1}$ (mainly C=O stretching of carboxylic acids), 1636 cm$^{-1}$
(mainly C=C stretching of aromatic rings and C=O stretching of conjugated carbonyl
groups) and 1390 cm$^{-1}$ (O-H deformation and C-O stretching of phenolic groups) were
observed (Fan et al., 2020; Fan et al., 2016b; Mukherjee et al., 2020; Wang et al., 2021a).
Additionally, weak peaks at 2929-2980 cm$^{-1}$ and 1045-1281 cm$^{-1}$, attributed to C-H
stretching of aliphatic -CH$_2$ and -CH$_3$, and C-O stretching of esters and ethers,
respectively, were also observed (Fan et al., 2016b; Wang et al., 2021a; Zhang et al.,
2021). These observations indicate that both HMW and LMW HULIS contain complex
multi-component mixtures of compounds, encompassing aliphatic and aromatic species,



as well as carboxyl and phenolic functional groups.

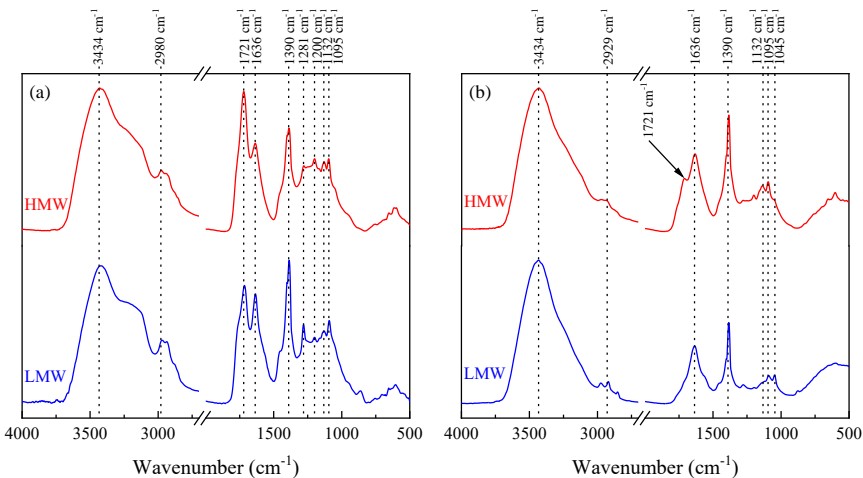


**Fig. 3.** FTIR spectra of HMW and LMW HULIS in (a) summer and (b) winter aerosols.

As shown in Fig. 3, more intense peaks at 1721 and 1636 cm$^{-1}$ were observed in
HULISs in summer aerosols compared to those in winter aerosols. In addition, the peaks
at 1045-1281 cm$^{-1}$ in summer HULISs appear to be more complex and overlapping than
those in winter HULISs. These findings imply higher abundances of aromatic carboxyl
acids and other O-containing groups (i.e., -OH, C=O and C-O) in summer HULISs than
in winter ones, possibly attributed to complex oxidation reactions prevailing in summer
season (Fan et al., 2020; Qin et al., 2022). This could be partly associated with the
enhanced oxidation processes driven by the higher concentration of O$_3$ in summer
(Table S1). Our previous study has proved that the O$_3$ oxidation of BB BrC lead to the
generation of more intense peaks at approximately 1725 cm$^{-1}$ (Fan et al., 2020).
Moreover, distinct differences in relative peak intensity between HMW and LMW
HULIS fractions were observed. HMW HULIS generally exhibit more intense at 1721



cm$^{-1}$ compared to LMW HULIS in both seasonal aerosols (Fig. 3). This finding suggests
that HMW HULIS contain a higher abundance of C=O groups, likely associated with
the oxidation of the unsaturated structures with addition of polar functional groups (e.g.,
-COOH, >C=O) during SOA processes (Fan et al., 2020; Pillar-Little and Guzman,

2018).


3.3 Molecular-level insights into MW HULIS
3.3.1. Seasonal variations in the molecular composition of MW HULIS
The molecular-level characteristics of MW HULIS were examined using negative
ESI- HR-MS analysis. Fig. 4 displays the reconstructed mass spectra of all HULIS
fraction in both summer and winter aerosols. Hundreds of peaks can be observed in the
spectra ranging from m/z 100 to 450 for all samples, with most ions being abundant
within the m/z 150-350 range. These spectrum characteristics are similar to those
previously reported for HULIS in ambient aerosols and BB emissions (He et al., 2023;
Song et al., 2022; Sun et al., 2021; Wang et al., 2019; Zhang et al., 2021; Zou et al.,

2023).



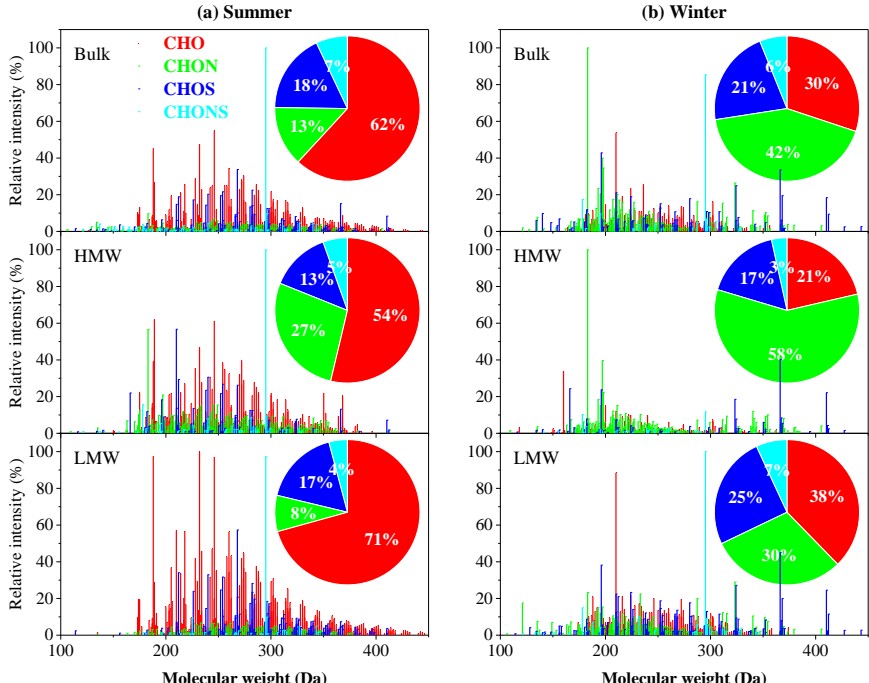

**Fig. 4.** Mass spectra of bulk and MW HULIS in (a) summer and (b) winter aerosols. The pie charts represent the intensity distributions of four compound categories (CHO, CHON, CHOS, and CHONS).

As listed in Table 2, the number of assigned formulas within MW HULIS in summer aerosols were 655-672, which was higher than the range of 470-506 observed in winter aerosols. This suggests that the MW HULIS in summer aerosols exhibited greater diversity than those in winter aerosols, mainly due to the stronger SOA formation that enhanced the heterogeneity of HULIS fractions in the summer. The identified formulas were then classified into four groups (i.e., CHO, CHON, CHOS and CHONS) according to their elemental composition. As depicted in pie charts in Fig. 4, summer HULISs are predominantly composed of CHO (54-71%), while winter





HULISs feature a high concentration of both CHON (30-58%) and CHO (21-38%). The
notably higher content of CHO in summer HULISs are likely due to a wide distribution
of biogenic VOC-derived SOAs during the summer season (Li et al., 2022; Sun et al.,
2023). CHON content in winter HULISs is generally higher than in summer ones,
potentially due to more significant contributions from direct BB, as well as secondary
nitrogen-related chemical processes during the winter season (He et al., 2023; Song et
al., 2022; Zhang et al., 2021; Zou et al., 2023). This finding is supported by the greater
number of fire spots (Fig. S1) and higher concentrations of $NO_2$ (Table S1) during
winter. The higher proportions of CHON compounds in aerosol HULIS typically lead
to enhanced light absorption capabilities (He et al., 2023; Song et al., 2022; Zeng et al.,
2021). This provides an strong explanation for why winter HULIS exhibit higher
$MAE_{365}$ values compared to summer HULIS. Additionally, CHOS is more abundant in
winter HULISs (17-25%) than in summer aerosols (13-18%). Previous studies have
demonstrated that both coal combustion and the oxidation initiated by $SO_2$ can lead to
the generation of larger amounts of S-containing compounds (Song et al., 2018; Song
et al., 2022; Zou et al., 2023). This finding suggested that the increased levels of coal
combustion and $SO_2$-related SOAs, as evidenced by higher concentration of $SO_2$ (Table
S1), are significant contributors to the presence of BrC in winter compared to in summer.





**Table 2.** The average values of intensity-weighted molecular weights (MW), elemental ratios, double bond equivalents (DBE), modified
aromaticity index ($AI_{mod}$) and carbon oxidation state ($OS_C$) for tentatively identified compounds of the bulk and MW HULIS samples.

| | | Elemental compositions | Number of formulas | $MW_w$ | $H/C_w$ | $O/C_w$ | $N/C_w$ | $S/C_w$ | $O/N_w$ | $O/S_w$ | $OM/OC_w$ | $DBE_w$ | $DBE/C_w$ | $DBE\text{-}O_w$ | $AI_{mod,w}$ | $OS_{C,w}$ |
|---|---|---|---|---|---|---|---|---|---|---|---|---|---|---|---|---|
| Summer | BULK | CHO | 376 | 276 | 1.39 | 0.59 | | | | | 1.91 | 4.69 | 0.39 | -2.20 | 0.15 | -0.21 |
| | | CHON | 270 | 260 | 1.35 | 0.60 | 0.15 | | 5.52 | | 2.10 | 4.99 | 0.50 | -1.17 | 0.27 | -0.14 |
| | | CHOS | 133 | 278 | 1.74 | 0.71 | | 0.11 | | 6.61 | 2.38 | 2.21 | 0.24 | -4.41 | 0.02 | -0.33 |
| | | CHONS | 81 | 265 | 1.69 | 0.59 | 0.18 | 0.14 | 4.98 | 5.20 | 2.49 | 3.38 | 0.36 | -1.95 | 0.09 | -0.52 |
| | | Total | 860 | 273 | 1.47 | 0.61 | 0.03 | 0.03 | 1.09 | 1.54 | 2.06 | 4.20 | 0.38 | -2.44 | 0.14 | -0.24 |
| | HMW | CHO | 270 | 264 | 1.41 | 0.55 | | | | | 1.85 | 4.51 | 0.38 | -1.70 | 0.16 | -0.32 |
| | | CHON | 264 | 248 | 1.42 | 0.61 | 0.14 | | 5.23 | | 2.09 | 4.44 | 0.47 | -1.33 | 0.24 | -0.20 |
| | | CHOS | 72 | 247 | 1.82 | 0.75 | | 0.14 | | 5.83 | 2.53 | 1.81 | 0.22 | -4.07 | 0.01 | -0.32 |
| | | CHONS | 39 | 269 | 1.55 | 0.64 | 0.19 | 0.15 | 4.87 | 5.16 | 2.60 | 3.82 | 0.44 | -1.65 | 0.14 | -0.28 |
| | | Total | 645 | 258 | 1.48 | 0.60 | 0.05 | 0.03 | 1.70 | 1.07 | 2.05 | 4.09 | 0.39 | -1.91 | 0.16 | -0.29 |
| | LMW | CHO | 365 | 275 | 1.34 | 0.64 | | | | | 1.97 | 4.82 | 0.42 | -2.47 | 0.16 | -0.05 |
| | | CHON | 155 | 272 | 1.39 | 0.66 | 0.11 | | 6.60 | | 2.12 | 4.80 | 0.45 | -2.09 | 0.18 | -0.08 |
| | | CHOS | 120 | 284 | 1.71 | 0.74 | | 0.11 | | 7.05 | 2.43 | 2.51 | 0.25 | -4.58 | 0.02 | -0.22 |
| | | CHONS | 32 | 322 | 1.71 | 0.64 | 0.11 | 0.11 | 6.59 | 6.63 | 2.42 | 3.32 | 0.30 | -3.41 | 0.06 | -0.42 |
| | | Total | 672 | 278 | 1.42 | 0.66 | 0.01 | 0.02 | 0.80 | 1.48 | 2.08 | 4.36 | 0.39 | -2.84 | 0.13 | -0.10 |
| Winter | BULK | CHO | 142 | 247 | 1.23 | 0.48 | | | | | 1.75 | 5.35 | 0.47 | -0.05 | 0.32 | -0.26 |
| | | CHON | 194 | 231 | 1.24 | 0.53 | 0.16 | | 3.96 | | 1.99 | 5.17 | 0.57 | 0.52 | 0.48 | -0.18 |
| | | CHOS | 79 | 271 | 1.93 | 0.51 | | 0.11 | | 4.97 | 2.14 | 1.21 | 0.14 | -3.80 | 0.04 | -0.91 |
| | | CHONS | 20 | 271 | 1.61 | 0.69 | 0.16 | 0.13 | 5.60 | 5.90 | 2.60 | 3.32 | 0.40 | -2.65 | 0.09 | -0.24 |
| | | Total | 435 | 247 | 1.41 | 0.52 | 0.08 | 0.03 | 2.02 | 1.42 | 1.99 | 4.27 | 0.44 | -0.77 | 0.31 | -0.37 |
| | HMW | CHO | 138 | 232 | 1.36 | 0.47 | | | | | 1.74 | 4.79 | 0.41 | 0.02 | 0.28 | -0.42 |



|  |  |  |  |  |  |  |  |  |  |  |  |  |  |  |
|---|---|---|---|---|---|---|---|---|---|---|---|---|---|---|
|  | CHON | 244 | 232 | 1.29 | 0.53 | 0.16 |  | 3.89 |  | 2.01 | 4.87 | 0.55 | 0.23 | 0.44 | -0.23 |
|  | CHOS | 59 | 292 | 2.08 | 0.40 |  | 0.12 |  | 4.40 | 2.04 | 0.46 | 0.06 | -4.06 | 0.02 | -1.27 |
|  | CHONS | 29 | 236 | 1.74 | 0.48 | 0.22 | 0.20 | 2.82 | 3.02 | 2.57 | 2.88 | 0.38 | -0.57 | 0.27 | -0.77 |
|  | Total | 470 | 242 | 1.46 | 0.50 | 0.10 | 0.03 | 2.37 | 0.85 | 1.98 | 4.04 | 0.43 | -0.57 | 0.33 | -0.46 |
| LMW | CHO | 176 | 249 | 1.23 | 0.54 |  |  |  |  | 1.82 | 5.25 | 0.48 | -0.65 | 0.30 | -0.15 |
|  | CHON | 195 | 239 | 1.34 | 0.49 | 0.16 |  | 3.99 |  | 1.95 | 4.88 | 0.52 | 0.29 | 0.45 | -0.36 |
|  | CHOS | 107 | 280 | 1.94 | 0.54 |  | 0.10 |  | 5.42 | 2.17 | 1.20 | 0.13 | -4.25 | 0.02 | -0.85 |
|  | CHONS | 28 | 272 | 1.67 | 0.69 | 0.16 | 0.14 | 5.89 | 6.03 | 2.63 | 2.99 | 0.37 | -3.15 | 0.13 | -0.29 |
|  | Total | 506 | 256 | 1.47 | 0.54 | 0.06 | 0.04 | 1.61 | 1.78 | 2.01 | 3.96 | 0.39 | -1.45 | 0.26 | -0.40 |





Table 2 summarizes the intensity-weighted molecular parameters for MW HULIS
in both summer and winter aerosols. Evidently, the MWw of summer HULISs are 258-
278, which are higher than the corresponding values of 242-256 for winter HULISs.
This observation indicates that summer HULISs exhibit larger sizes, consistent with
their higher HPSEC-derived Mw and Mn compared to winter HULISs. Moreover,
summer HULISs exhibit higher O/Cw ranging from 0.60 to 0.66, as well as $OS_{C,w}$
ranging from -0.29 to -0.10, which exceed the respective values of 0.50 to 0.54 and -
0.46 to -0.37 observed in winter HULISs. Conversely, winter HULISs display higher
$AI_{mod,w}$ values (0.26-0.33) than those (0.13-0.16) for summer ones. These findings
suggest that summer HULISs are characterized by a high degree of oxidation, while
winter HULISs exhibit stronger aromaticity.

3.3.2. Comparison on molecular composition of HMW and LMW HULIS
**CHO compounds.** The CHO compounds are prominent constituents within
HULIS fractions, accounting for 54% and 21% in summer and winter HMW HULIS,
respectively, whereas these proportions increase to 71% and 38% in LMW HULIS (Fig.
4). It is worth noting that CHO compounds that undergo deprotonation in ESI- mode
are likely associated with the presence of carboxyl, carbonyl, alcohol and ester (Lin et
al., 2012; Wang et al., 2018). Moreover, CHO compounds in LMW HULIS exhibit a
higher oxygenation level compared to HMW HULIS, as evidenced by the higher $O/C_w$
and $OS_{C,w}$ values. As shown in Table 2, the $O/C_w$ for CHO in LMW HULIS are 0.55-
0.64, which are higher than 0.47-0.54 observed in HMW HULIS. In contrast, the $H/Cw$



for CHO in HMW HULIS were consistently higher than those in LMW HULIS, with
values of 1.41 vs. 1.34 in summer and 1.36 vs. 1.23 in winter (Table 2). This disparity
strongly suggests a higher saturation level of CHO compounds within HMW HULIS.
This conclusion is further corroborated by the lower $DBE_w$ and $AI_{mod,w}$ observed for
CHO in HMW HULIS compared to LMW HULIS (Table 2). It is known that these
values serve as estimations of C=C density and aromatic and condensed aromatic
structures (Song et al., 2022; Zhang et al., 2021). Taken together, the CHO compounds
within HMW HULIS exhibit a more aliphatic nature but lower aromaticity and
oxidation levels when compared to those within LMW HULIS.
**CHON compounds.** HMW HULIS fractions consist of a higher proportion of
CHON compounds compared to LMW HULIS, with proportions of 27% vs. 8% in
summer and 58% vs. 30% in winter (Fig. 4). This observation suggests that HMW
HULIS contain a higher content of N-containing components. It is noted that the LMW
HULIS are generally characterized by higher $O/N_w$ values of 6.60 in summer and 3.99
in winter compared to 5.23 in summer and 3.89 in winter for HMW HULIS. This
indicates that the CHON compounds within LMW HULIS are more highly oxidized
than those within HMW HULIS. In general, compounds with O/N ≥ 3 are indicative of
oxidized N groups such as nitro ($-NO_2$) or nitrooxy ($-ONO_2$), while compounds with
O/N < 3 may denote the reduced N-containing functional groups (i.e., amines) (He et
al., 2023; Song et al., 2022; Zeng et al., 2021). In this study, a majority of the CHON
compounds, comprising 73-85% in summer and 59-64% in winter, exhibited O/N ≥ 3
in both MW HULIS fractions. This suggests that high concentrations of nitro



compounds or organonitrates dominate the CHON compounds (Sun et al., 2023; Wang
et al., 2018; Zeng et al., 2021), especially in summer samples, primarily due to the
hydroxyl radical oxidation of biogenic or anthropogenic VOC precursors, as well as
BB emissions (Song et al., 2022; Sun et al., 2021; Zhang et al., 2021; Zou et al., 2023).
Furthermore, the CHON compounds exhibiting O/N ≥ 3 were more abundant in LMW
HULIS compared to HMW HULIS, accounting for 85% vs. 73% in summer and 64%
vs. 59% in winter. In contrast, HMW HULIS contained more CHON compounds with
O/N < 3 compared to LMW HULIS. These findings collectively indicate that the CHON
within HMW HULIS possess lower content of nitro compounds or organonitrates than
LMW HULIS. Based on FTIR analysis, it is known that HMW HULIS contain more
carboxylic groups than LMW HULIS, which indicate a higher likelihood of HMW
HULIS containing more amino acids.
**CHOS and CHONS compounds.** In this study, we observed that CHOS
accounted for proportions of 13% to 25% in all MW HULIS fractions, while CHONS
had a lower proportion of 3% to 7% (Fig. 4). Notably, the distribution of CHOS differed
between HMW and LMW HULIS in both season samples. As depicted in Fig. 4, HMW
HULIS contained fewer CHOS compounds compared to LMW HULIS, with
proportions of 13% vs. 17% in summer and 17% vs. 25% in winter. This finding
suggests that a greater number of CHOS compounds are incorporated into the LMW
HULIS fractions, which potentially leading to a reduction in the light absorption of
LMW HULIS (Zeng et al., 2021; Zhang et al., 2021). Furthermore, as indicated in Table
2, both the CHOS and CHONS within LMW HULIS exhibited higher O/Sw values than



HMW HULIS in both seasonal samples. Consequently, the S-containing compounds
within LMW HULIS were characterized by a higher degree of oxidation, primarily
attributed to $SO_2$-related chemical oxidation process, in comparison to those in HMW
HULIS. Moreover, it was observed that 61% to 92% of CHOS compounds exhibited
O/S > 4, and 3% to 43% of CHONS compounds with O/S > 7 for all MW HULIS
fractions. Among them, HMW HULIS own lower proportions of CHOS with O/S > 4
and CHONS with O/S > 7 than LMW HULIS, suggesting a reduced presence of
potential organosulfates and nitrooxyorganosulfates within HMW HULIS (Sun et al.,
2023; Wang et al., 2018; Zeng et al., 2021; Zou et al., 2023).

3.3.3. Comparative analysis of unique molecular formulas in HMW and LMW HULIS
In this study, particular emphasis was placed on the unique molecular formulas
within the HMW or LMW HULIS fractions. Fig. 5a, b illustrates the Van Krevelen (VK)
diagram depicting the distribution of unique molecular formulas within HMW and
LMW HULIS in summer and winter samples. It is evident that a majority of unique
formulas within LMW HULIS are concentrated around the origin with O/C > 0.5,
accounting for 83% in summer and 64% in winter. In contrast, most formulas within
HMW HULIS exhibited O/C < 0.5, representing about 58% for both seasonal samples.
These findings indicate that the unique molecules within LMW HULIS consist of more
polar O-containing organic compounds than those within HMW HULIS.



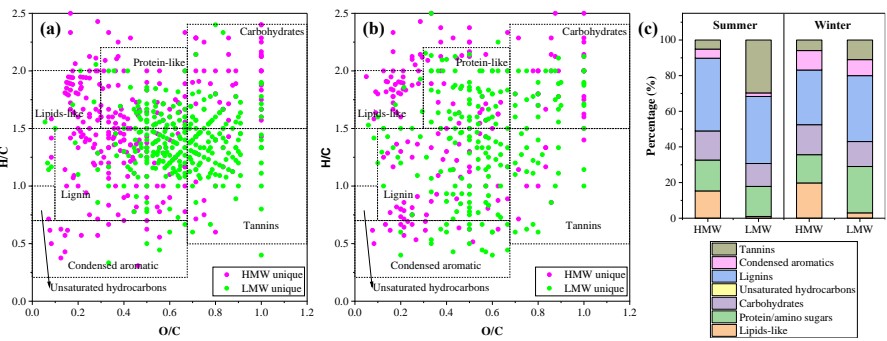

**Fig. 5.** Van Krevelen diagrams for the unique molecular formulas within HMW and LMW HULIS from (a) summer and (b) winter aerosols. (c) The contributions of major substances classes in unique formulas.

The molecular formulas are further categorized into seven groups based on previous studies, including lignin-like species, protein/amino sugars, condensed aromatics, tannin-like species, carbohydrate-like species, unsaturated hydrocarbons, and lipid-like species (He et al., 2023; Sun et al., 2021; Sun et al., 2023). The classification rules for these formulas can be found in Table S2. Fig. 5c provides an overview of the relative contributions of the number of unique formulas from each of the seven groups for HMW and LMW HULIS. The results indicate that the dominant substance class in the unique formulas within both MW HULIS are lignin-like species, accounting for proportions of 31-40%. This finding indicates that lignin derivatives are fundamental components in both HMW and LMW HULIS either in summer or winter aerosols. Additionally, there are notable differences in the molecular characteristics of lignin-like species within HMW and LMW HULIS. As listed in Table S3, lignin-like species within HMW HULIS exhibit lower MWw and O/Cw, but higher N/Cw and AImod,w values than those within LMW HULIS in both seasonal samples. These



observations suggest that the unique lignin-like substances in HMW HULIS likely
contain more N-enriched and highly aromatic species, while those in LMW HULIS
tend to concentrate more aliphatic O-containing compounds. These distinctions in
composition and characteristics between HMW and LMW HULIS fractions provide
valuable insights into their origins and transformations in the atmosphere.
Moreover, there are notable variations in the contributions of lipids-like,
protein/amino sugars, carbohydrates, condensed aromatics, and tannins species
between HMW and LMW HULIS. In general, HMW HULIS have a higher proportion
of lipids-like species, carbohydrates and condensed aromatics than LMW HULIS in
both summer and winter aerosols. Among these, the most remarkable difference in
composition between HMW HULIS and LMW HULIS is seen in lipids-like species,
accounting for 15% versus 1% in summer and 20% versus 3% in winter (Fig. 5). As
reported in previous studies, lipids-like species primarily originate from biogenic
emissions (He et al., 2023; Li et al., 2022; Sun et al., 2021). This suggests that there is
a stronger contribution from biogenic emissions to HMW HULIS. Additionally, these
species in HMW HULIS were usually characterized by lower $DBE_w$ and slightly lower
$OS_{C,w}$ when compared to LMW HULIS (Table S3), indicating they present stronger
saturation and fewer oxidized substituents. On the other hand, tannins species
contribute a higher proportion to LMW HULIS, constituting 30% in summer and 11%
in winter, while comprising only 5%-6% in HMW HULIS in both season aerosols.
Tannin-like species are known to consist of various polyphenolic groups containing
hydroxyl and carboxylic functional groups (He et al., 2023; Li et al., 2022; Ning et al.,



2019; Sun et al., 2021). The slightly lower DBEw but much higher DBE-Ow for unique
tannin-like species within HMW HULIS were observed compared to LMW HULIS
(Table S3), suggesting that the former ones are enriched in more unsaturated O-
containing functional groups, particularly carboxylic functional groups.

3.4. Atmospheric implications

This study provides comprehensive comparison between HMW and LMW HULIS

regarding their distributions, chemical structures, molecular sizes and compositions.
HMW HULIS appear to be larger than LMW HULIS, as evidenced by both
ultrafiltration natures and the MW distributions of chromophores analyzed by HPSEC.
However, HRMS analysis revealed that the average MWw of identified formulas within
HMW HULIS were lower than those of LMW HULIS (Table 2). This discrepancy can
likely be attributed to the "assembled structures" that construct the aerosol HULIS, as
suggested in many previous studies focusing on HULIS and BrC characterization (Fan
et al., 2021; Fan et al., 2023; Phillips et al., 2017; Qin et al., 2022). In fact, the results
from EEM-FRI and FTIR analysis support the notion that HMW and LMW HULIS
likely consist of potential structures assembled by similar basic fluorophores and
functional groups. Based on this theory, HMW HULIS may consist of macromolecular
species primarily assembled from small molecules through weak forces (i.e., π–π, van
der Waals, hydrophobic, or hydrogen bonds) and/or charge-transfer interactions (Fan et
al., 2021; Phillips et al., 2017), which can potentially disassemble during ESI ionization
and form low MW molecules.



Based on the molecular-level characterization, significant distinctions in
properties between HMW HULIS and LMW HULIS become evident. HMW HULIS
generally exhibit stronger aromaticity but lower oxidation degree when compared to
LMW HULIS. In terms of molecular composition, HMW HULIS contain higher
quantities of CHON species but lower quantities of CHO compounds than LMW
HULIS. Furthermore, more lipids-like species were identified as unique molecules in
HMW HULIS, while more tannin-like species with abundant carboxylic groups were
observed as unique molecules in LMW HULIS. Given these pronounced differences
between HMW and LMW HULIS, it can be speculated that the higher levels of aromatic
structures, greater presence of CHON molecules and the presence of lipids-like species
may serve as driving factors in the formation of potential assembled structures in HMW
HULIS. Additionally, it is well-established that CHON can enhance the light absorption
of organic aerosols (OA), while CHO species may have the opposite effect, weakening
light absorption (He et al., 2023; Song et al., 2022; Wang et al., 2019; Zeng et al., 2021).
Therefore, it is reasonable to conclude that HMW HULIS possess stronger light
absorbing capability, which is consistent with their larger $MAE_{365}$ values.
Importantly, HMW HULIS contain amounts of carboxylic functional groups,
reduced nitrogen species (e.g., amines) and aromatic species than LMW HULIS. These
functional groups have strong complexation abilities with transition metals (Wang et
al., 2021a; Wang et al., 2021b), thus influencing the transformation and chemical
behavior of metals. Moreover, the OA-metals complex can potentially enhance the
catalytic generation of reactive oxygen species (ROS) in organic aerosols (Win et al.,



2018; Zhang et al., 2022a), thereby playing significant roles in adverse health effects of
OA. These results reinforce the significance of HMW HULIS in light absorption, metal
complexation, and the potential ROS generation ability of aerosol BrC.

**4. Conclusions**

This study successfully isolated and characterized HMW and LMW HULIS in

atmospheric aerosols using the UF-SPE technique, yielding insights into their
distribution, optical properties and molecular-level characteristics. Both HMW and
LMW HULIS exhibited a continuum of MW distributions ranging from 100 to 20,000
Da. However, HMW HULIS displayed more extensive and intricate MW distributions,
suggesting differences in their sources and formation processes compared to LMW
HULIS. In general, HMW HULIS constituted a higher percentage of TOC and UV254
in aerosols compared to LMW HULIS, indicating the prevalence of hydrophobic and
conjugated aromatic structures in the former. Moreover, HMW HULIS exhibited higher
aromaticity, stronger light absorption abilities, weaker spectra dependence, and stronger
humification and conjugation, compared to LMW HULIS. Interestingly, HRMS
analysis revealed slightly lower MWw values for HMW HULIS than LMW HULIS,
which contradicted the HPSEC results and the nature of UF fractionation. This finding
strongly suggests the possibility of small molecules assembling to form
macromolecules in HMW HULIS. Regarding molecular composition, HMW HULIS
contained a higer proportion of CHON compounds, constituting 27% vs. 8% in summer
and 58% vs. 30% in winter, while LMW HULIS were primarily composed of CHO



compounds, accounting for 71% vs. 54%% in summer and 38% vs. 21% in winter. Both
HMW and LMW HULIS featured lignin-like substances as major unique molecular
formulas, but HMW HULIS exhibited more N-enriched and highly aromatic species,
whereas LMW HULIS contained a higher proportion of polar O-containing functional
groups. Additionally, HMW HULIS included a greater number of unique lipids-like
compounds, while LMW HULIS tend to concentrate more tannin-like compounds.
These observations shed light on the complex nature of MW HULIS, and their diverse
sources and transformations. Future research should expand the geographical and
seasonal coverage to gain a more comprehensive understanding of the molecular-level
characteristics of MW HULIS in various atmospheric environments. Furthermore,
exploring additional physicochemical properties of MW HULIS will provide valuable
insights into their potential health and environmental implications. Overall, this study
offers valuable insights into the molecular-level characteristics of aerosol HULIS,
enhancing our understanding of their evolution, sources and potential environmental
effects.
**Author contribution**
**Xingjun Fan:** Methodology, Supervision, Funding acquisition, Writing-review &
editing. **Ao Cheng:** Sampling, Data curation. **Xufang Yu:** Writing-review & editing.
**Tao Cao:** Sampling, Investigation. **Dan Chen:** Investigation, Data curation. **Wenchao
Ji:** Formal analysis. **Yongbing Cai:** Writing-review & editing. **Fande Meng:** Writing-
review & editing. **Jianzhong Song:** Methodology, Writing-review & editing. **Pingan
Peng:** Writing-review & editing.



**Declaration of Competing Interest**

The authors declare that they have no known competing financial interests or personal relationships that could have appeared to influence the work reported in this paper.

**Acknowledgments**

This study was supported by the Natural Science Foundation of China (42192514, 52100114), the Anhui Provincial Natural Science Foundation (2108085MD140, 2108085QB56), and the State key Laboratory of Organic Geochemistry, GIGCAS (SKLOG202101), Anhui Provincial Key Science Foundation for Outstanding Young Talent (2022AH030145, gxyqZD2021126).

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

Transformations of Humic-Like Substances and Further Cognitions Revealed by Optical
Methods. Environ. Sci. Technol. 56, 7578-7587.
Qin, J., Zhang, L., Zhou, X., Duan, J., Mu, S., Xiao, K., Hu, J., Tan, J., 2018. Fluorescence fingerprinting
properties for exploring water-soluble organic compounds in PM 2.5 in an industrial city of
northwest China. Atmos. Environ. 184, 203-211.
Santos, P.S.M., Otero, M., Duarte, R.M.B.O., Duarte, A.C., 2009. Spectroscopic characterization of
dissolved organic matter isolated from rainwater. Chemosphere 74, 1053-1061.
Santos, P.S.M., Santos, E.B.H., Duarte, A.C., 2012. First spectroscopic study on the structural features
of dissolved organic matter isolated from rainwater in different seasons. Sci. Total Environ. 426,
172-179.

Song, J., Li, M., Jiang, B., Wei, S., Fan, X., Peng, P., 2018. Molecular Characterization of Water-Soluble
Humic like Substances in Smoke Particles Emitted from Combustion of Biomass Materials and
Coal Using Ultrahigh-Resolution Electrospray Ionization Fourier Transform Ion Cyclotron
Resonance Mass Spectrometry. Environ. Sci. Technol. 52, 2575-2585.
Song, J., Li, M., Zou, C., Cao, T., Fan, X., Jiang, B., Yu, Z., Jia, W., Peng, P.a., 2022. Molecular
Characterization of Nitrogen-Containing Compounds in Humic-like Substances Emitted from
Biomass Burning and Coal Combustion. Environ. Sci. Technol. 56, 119-130.



Song, J.Z., Huang, W.L., Peng, P.A., Xiao, B.H., Ma, Y.J., 2010. Humic Acid Molecular Weight
Estimation by High-Performance Size-Exclusion Chromatography with Ultraviolet Absorbance
Detection and Refractive Index Detection. Soil. Sci. Soc. Am. J. 74, 2013-2020.

Sun, H., Li, X., Zhu, C., Huo, Y., Zhu, Z., Wei, Y., Yao, L., Xiao, H., Chen, J., 2021. Molecular
composition and optical property of humic-like substances (HULIS) in winter-time PM2.5 in
the rural area of North China Plain. Atmos. Environ. 252, 118316.

Sun, H., Wu, Z., Kang, X., Zhu, C., Yu, L., Li, R., Lin, Z., Chen, J., 2023. Molecular characterization of
humic-like substances (HULIS) in atmospheric particles (PM2.5) in offshore Eastern China Sea
(OECS) using solid-phase extraction coupled with ESI FT-ICR MS. Atmos. Environ. 294,
119523.

Tang, J., Li, J., Mo, Y., Safaei Khorram, M., Chen, Y., Tang, J., Zhang, Y., Song, J., Zhang, G., 2020.
Light absorption and emissions inventory of humic-like substances from simulated rainforest
biomass burning in Southeast Asia. Environmental Pollution 262, 114266.

Wang, K., Zhang, Y., Huang, R.-J., Cao, J., Hoffmann, T., 2018. UHPLC-Orbitrap mass spectrometric
characterization of organic aerosol from a central European city (Mainz, Germany) and a
Chinese megacity (Beijing). Atmos. Environ. 189, 22-29.

Wang, X., Qin, Y., Qin, J., Long, X., Qi, T., Chen, R., Xiao, K., Tan, J., 2021a. Spectroscopic insight into
the pH-dependent interactions between atmospheric heavy metals (Cu and Zn) and water-
soluble organic compounds in PM2.5. Sci. Total Environ. 767, 145261.

Wang, X.B., Qin, Y.Y., Qin, J.J., Yang, Y.R., Qi, T., Chen, R.Z., Tan, J.H., Xiao, K., 2021b. The interaction
laws of atmospheric heavy metal ions and water-soluble organic compounds in PM2.5 based on
the excitation-emission matrix fluorescence spectroscopy. Journal of Hazardous Materials 402,
8.

Wang, Y., Hu, M., Lin, P., Tan, T., Li, M., Xu, N., Zheng, J., Du, Z., Qin, Y., Wu, Y., Lu, S., Song, Y., Wu,
Z., Guo, S., Zeng, L., Huang, X., He, L., 2019. Enhancement in Particulate Organic Nitrogen
and Light Absorption of Humic-Like Substances over Tibetan Plateau Due to Long-Range
Transported Biomass Burning Emissions. Environ. Sci. Technol. 53, 14222-14232.

Win, M.S., Tian, Z., Zhao, H., Xiao, K., Peng, J., Shang, Y., Wu, M., Xiu, G., Lu, S., Yonemochi, S.,
Wang, Q., 2018. Atmospheric HULIS and its ability to mediate the reactive oxygen species
(ROS): A review. J Environ Sci (China) 71, 13-31.

Wong, J.P.S., Nenes, A., Weber, R.J., 2017. Changes in Light Absorptivity of Molecular Weight
Separated Brown Carbon Due to Photolytic Aging. Environ. Sci. Technol. 51, 8414-8421.

Wong, J.P.S., Tsagkaraki, M., Tsiodra, I., Mihalopoulos, N., Violaki, K., Kanakidou, M., Sciare, J., Nenes,
817              A., Weber, R.J., 2019. Atmospheric evolution of molecular-weight-separated brown carbon
from biomass burning. Atmos. Chem. Phys. 19, 7319-7334.

Wu, G., Fu, P., Ram, K., Song, J., Chen, Q., Kawamura, K., Wan, X., Kang, S., Wang, X., Laskin, A.,
Cong, Z., 2021. Fluorescence characteristics of water-soluble organic carbon in atmospheric
aerosol☆. Environmental Pollution 268, 115906.

Wu, G., Wan, X., Gao, S., Fu, P., Yin, Y., Li, G., Zhang, G., Kang, S., Ram, K., Cong, Z., 2018. Humic-
Like Substances (HULIS) in Aerosols of Central Tibetan Plateau (Nam Co, 4730 m asl):
Abundance, Light Absorption Properties, and Sources. Environ. Sci. Technol. 52, 7203-7211.

Wu, G., Wan, X., Ram, K., Li, P., Liu, B., Yin, Y., Fu, P., Loewen, M., Gao, S., Kang, S., Kawamura, K.,
Wang, Y., Cong, Z., 2020. Light absorption, fluorescence properties and sources of brown
carbon aerosols in the Southeast Tibetan Plateau. Environmental Pollution 257, 113616.



Yuan, W., Huang, R.-J., Yang, L., Ni, H., Wang, T., Cao, W., Duan, J., Guo, J., Huang, H., Hoffmann, T.,
2021. Concentrations, optical properties and sources of humic-like substances (HULIS) in fine
particulate matter in Xi'an, Northwest China. Sci. Total Environ. 789, 147902.
Zeng, Y., Ning, Y., Shen, Z., Zhang, L., Zhang, T., Lei, Y., Zhang, Q., Li, G., Xu, H., Ho, S.S.H., Cao, J.,
2021. The Roles of N, S, and O in Molecular Absorption Features of Brown Carbon in PM2.5
in a Typical Semi-Arid Megacity in Northwestern China. Journal of Geophysical Research:
Atmospheres 126, e2021JD034791.
Zhang, T., Huang, S., Wang, D., Sun, J., Zhang, Q., Xu, H., Hang Ho, S.S., Cao, J., Shen, Z., 2022a.
Seasonal and diurnal variation of PM2.5 HULIS over Xi'an in Northwest China: Optical
properties, chemical functional group, and relationship with reactive oxygen species (ROS).
Atmos. Environ. 268, 118782.
Zhang, T., Shen, Z., Huang, S., Lei, Y., Zeng, Y., Sun, J., Zhang, Q., Ho, S.S.H., Xu, H., Cao, J., 2022b.
Optical properties, molecular characterizations, and oxidative potentials of different polarity
levels of water-soluble organic matters in winter PM2.5 in six China's megacities. Sci. Total
Environ. 853, 158600.
Zhang, T., Shen, Z., Zeng, Y., Cheng, C., Wang, D., Zhang, Q., Lei, Y., Zhang, Y., Sun, J., Xu, H., Ho,
S.S.H., Cao, J., 2021. Light absorption properties and molecular profiles of HULIS in PM2.5
emitted from biomass burning in traditional "Heated Kang" in Northwest China. Sci. Total
Environ. 776, 146014.
Zhang, T., Shen, Z., Zhang, L., Tang, Z., Zhang, Q., Chen, Q., Lei, Y., Zeng, Y., Xu, H., Cao, J., 2020.
PM2.5 Humic-like substances over Xi'an, China: Optical properties, chemical functional group,
and source identification. Atmos. Res. 234, 104784.
Zhang, W., Li, L., Wang, D., Wang, R., Yu, S., Gao, N., 2022c. Characterizing dissolved organic matter
in aquatic environments by size exclusion chromatography coupled with multiple detectors.
Anal. Chim. Acta 1191, 339358.
Zheng, G.J., He, K.B., Duan, F.K., Cheng, Y., Ma, Y.L., 2013. Measurement of humic-like substances in
aerosols: A review. Environmental Pollution 181, 301-314.
Zou, C., Cao, T., Li, M., Song, J., Jiang, B., Jia, W., Li, J., Ding, X., Yu, Z., Zhang, G., Peng, P.a., 2023.
Measurement report: Changes in light absorption and molecular composition of water-soluble
humic-like substances during a winter haze bloom-decay process in Guangzhou, China. Atmos.
Chem. Phys. 23, 963-979.
Zou, C., Li, M., Cao, T., Zhu, M., Fan, X., Peng, S., Song, J., Jiang, B., Jia, W., Yu, C., Song, H., Yu, Z.,
Li, J., Zhang, G., Peng, P.a., 2020. Comparison of solid phase extraction methods for the
measurement of humic-like substances (HULIS) in atmospheric particles. Atmos. Environ. 225,
117370.

