# Peer review of "Distribution, chemical and molecular composition of high and"

_EGUsphere, 2023_

## Author Comment (AC1)

**Response to Editors and Reviewers**

Thanks for reviewers' careful reading and constructive comments and suggestions. We made every effort to respond to reviewers' questions point to point, and carefully revised our manuscript and Supporting Information based on their comments. To enhance clarity, we present the reviewers' comments in regular black font, while our responses are displayed in blue normal font. The modified content in both the manuscript and the Supporting Information is highlighted in red font.

**Anonymous Referee #2**

The authors characterized High (HMW) and Low Molecular Weight (LMW) using a combination of analytical techniques to differentiate the molecular, functional and optical properties between samples collected during summer and winter in Anhui (China). The manuscript is well written and the link between seasonal sources, HULIS functional and molecular level of information are well linked with the optical properties. The results are a bit long in some sections, and perhaps some of the material (less novel results) could be moved to the Supplementary information to avoid losing the readers and keep the manuscript clear and concise. I would recommend this manuscript for publication after addressing the major comments.

[Response]: Thanks for the reviewer's comment. The contents on FTIR analysis and results have been removed into supporting information.

General comments:

In section 2.6, are there any biases due to the use of the negative mode with ESI-MS? Can the authors estimate the fraction of HMW and LMW HULIS lost during the ESI characterization, or do they assume that most of the HMW HULIS will end up in a multiple charge state or "disassemble" (based on the statement page 32 line 596)? The idea of HMW HULIS being aggregates of smaller molecules should be developed in more details throughout the discussion instead of only mentioning it toward the end.

[Response]: Thanks for the reviewer's comment. Yes, biases can arise due to the use of the negative mode with ESI- HRMS. Electrospray ionization (ESI) is a form of soft ionization source where compounds with high molecular weight and poor stability won't decompose during the ionization process. Simultaneously, it's a selective ionization source primarily suitable for highly polar compounds containing functional groups prone to losing protons, making them more effectively detectable in the negative ESI mode (Lin et al., 2012; Song et al., 2018, 2022). Consequently, in the negative ESI mode, the detection of low-polarity substances like polycyclic aromatic hydrocarbons, aliphatic hydrocarbons, may be missed, and substances easily protonated, such as nitrogenous substances, may also be absent (He et al., 2023; Zou et al., 2023). In general, the ESI- HRMS can reveal molecular composition of a subset of organic molecules that are preferentially ionized in the negative ESI source, rather than representing the entire HULIS composition. Regarding the fraction of HULIS lost during ESI characterization, it's beyond our analysis scope due to the limitations of ESI HRMS. The concept of HMW HULIS being aggregates of small molecules has been elaborated with more details in the revised manuscript.

[Revise]: The relevant statements on biases of ESI HRMS have been added in lines 359-392, 437-439 in the revised manuscript:

"It is worth noting that the ESI- HRMS could reveal molecular composition of a subset of organic molecules that are biased ionized in the negative ESI source rather than representing the entire HULIS composition (He et al., 2023; Lin et al., 2012)."

"This difference could be explained by several factors: (1) ESI- HRMS is based towards relatively small molecules that easily protonated in negative ESI mode (He et al., 2023; Lin et al., 2012)"

The content on assembly of MW HULIS has been incorporated in lines 244-246, 356-359, 441-443 in the revised manuscript:

"In addition, the magnitude peak at 570 Da (peak i) in HMW HULIS may indicate the incorporation of small molecule through weak interactions based on $\pi$–$\pi$ and/or van der Waals forces between the HULIS components (Fan et al., 2021; Piccolo, 2002)."

"The findings imply an assembly of small and heterogeneous molecules to form bulk

HULIS through weak intramolecular forces (i.e., π–π, van der Waals, hydrophobic, or hydrogen bonds) (Fan et al., 2021; Piccolo, 2002) and/or charge-transfer interactions (Phillips et al., 2017; Qin et al., 2022)."

"(3) the potential disassembly of larger molecules stabilized by weak forces during electrospray ionization of HRMS (Fan et al., 2021; Phillips et al., 2017)."

[Reference]:

(1)  He, T., Wu, Y., Wang, D., Cai, J., Song, J., Yu, Z., Zeng, X., Peng, P.a., 2023. Molecular compositions and optical properties of water-soluble brown carbon during the autumn and winter in Guangzhou, China. Atmos. Environ. 296, 119573.

(2)  Lin, P., Rincon, A.G., Kalberer, M., Yu, J.Z., 2012. Elemental composition of HULIS in the Pearl River Delta Region, China: results inferred from positive and negative electrospray high resolution mass spectrometric data. Environ. Sci. Technol. 46, 7454-7462.

(3)  Song, J., Li, M., Jiang, B., Wei, S., Fan, X., Peng, P., 2018. Molecular Characterization of Water-Soluble Humic like Substances in Smoke Particles Emitted from Combustion of Biomass Materials and Coal Using Ultrahigh-Resolution Electrospray Ionization Fourier Transform Ion Cyclotron Resonance Mass Spectrometry. Environ. Sci. Technol. 52, 2575-2585.

(4)  Song, J., Li, M., Zou, C., Cao, T., Fan, X., Jiang, B., Yu, Z., Jia, W., Peng, P.a., 2022. Molecular Characterization of Nitrogen-Containing Compounds in Humic-like Substances Emitted from Biomass Burning and Coal Combustion. Environ. Sci. Technol. 56, 119-130.

(5)  Zou, C., Cao, T., Li, M., Song, J., Jiang, B., Jia, W., Li, J., Ding, X., Yu, Z., Zhang, G., Peng, P.a., 2023. Measurement report: Changes in light absorption and molecular composition of water-soluble humic-like substances during a winter haze bloom-decay process in Guangzhou, China. Atmospheric Chemistry and Physics 23, 963-979.

Would the contribution of small molecules at 570Da (peak i) present in HMW chromatogram (similarly the fraction of HMW in the LMW, peak ii ~2200 Da) as seen in Figure 1, influence the subsequent observation, in particular the FTIR spectra?

[Response]: Thanks for the reviewer's comments. We admit that there might be a small number of small molecules in HMW HULIS and a small number of large molecules in LWM HULIS, primarily due to the following reasons: (1) Large size molecules obtained by ultrafiltration may incorporate small molecules as a result of weak interactions based on π–π and/or van der Waals forces between the HULIS components (Fan et al., 2021; Piccolo, 2002). These small molecules can be released in the high-pressure mobile phase and thus detected in the HMW fraction; (2) For small size molecules obtained by ultrafiltration, some long-chain substances may pass through the ultrafiltration membrane under pressure during the ultrafiltration process and can be detected in the LMW fraction (Fan et al., 2021; Lee et al., 2020; Wang et al., 2018). Overall, for comprehensive analysis methods such as EEM, HPSEC, FTIR and HRMS, the HMW and LMW HULIS present distinct molecular characteristics, rendering the effects of these minor components negligible.

[Reference]
(1) Fan, X., Cai, F., Xu, C., Yu, X., Wang, Y., Xiao, X., Ji, W., Cao, T., Song, J., Peng, P.a., 2021. Molecular weight-dependent abundance, absorption, and fluorescence characteristics of water-soluble organic matter in atmospheric aerosols. Atmos. Environ. 247.

(2) Lee, Y.K., Romera-Castillo, C., Hong, S., Hur, J., 2020. Characteristics of microplastic polymer-derived dissolved organic matter and its potential as a disinfection byproduct precursor. Water Res. 175, 115678.

(3) Piccolo, A., 2002. The supramolecular structure of humic substances: A novel understanding of humus chemistry and implications in soil science, Advances in Agronomy. Academic Press, pp. 57-134.

(4) Wang, K., Zhang, Y., Huang, R.-J., Cao, J., Hoffmann, T., 2018. UHPLC-Orbitrap mass spectrometric characterization of organic aerosol from a central European

city (Mainz, Germany) and a Chinese megacity (Beijing). Atmos. Environ. 189, 22-29.

Could Tar Balls influence the samples and absorption measurement?

[Response]: Tar balls are a specific type of particle produced from wood combustion, especially during biomass smoldering burning, and are abundant in the troposphere (Chen et al., 2017; Li et al., 2019). Microanalysis has revealed that tar balls are homogeneous spherical carbonaceous particles with sizes ranging from tens to hundreds nanometers. However, during their long atmospheric transport, tar balls particles occasionally form aggregates up to ten particles, including coagulation with dust particles (Hand, et al., 2005, Tóth et al., 2014). Importantly, tar balls are generally considered as water-insoluble brown carbon (Corbin et al., 2019). In the current study, the focus is on water-soluble HULIS, which allows the avoidance of the influence of tar balls.

[Reference]:

(1) Chen, J., Li, C., Ristovski, Z., Milic, A., Gu, Y., Islam, M. S., Wang, S., Hao, J., Zhang, H., and He, C.: A review of biomass burning: Emissions and impacts on air quality, health and climate in China, Sci. Total Environ., 579, 1000 – 1034, https://doi.org/10.1016/j.scitotenv.2016.11.025, 2017.

(2) Corbin, J. C. and Gysel-Beer, M.: Detection of tar brown carbon with a single particle soot photometer (SP2), Atmos. Chem. Phys., 19, 15673 – 15690, https://doi.org/10.5194/acp-19-15673-2019, 2019.

(3) Hand, J. L., Malm, W., Laskin, A., Day, D., Lee, T. B., Wang, C., Carrico, C., Carrillo, J., Cowin, J. P., and Collett, J.: Optical, physical, and chemical properties of tar balls observed during the Yosemite Aerosol Characterization Study, J. Geophys. Res.- Atmos., 110, D21210, https://doi.org/10.1029/2004JD005728, 2005

(4) Li, C., He, Q., Schade, J., Passig, J., Zimmermann, R., Meidan, D., Laskin, A., and Rudich, Y.: Dynamic changes in optical and chemical properties of tar ball

aerosols by atmospheric photochemical aging, Atmos. Chem. Phys., 19, 139 – 163, https://doi.org/10.5194/acp-19-139-2019, 2019.

(5) Tóth, A., Hoffer, A., Nyiro-Kósa, I., Pósfai, M., and Gelencsér, A.: Atmospheric tar balls: aged primary droplets from biomass burning?, Atmos. Chem. Phys., 14, 6669 – 6675, https://doi.org/10.5194/acp-14-6669-2014, 2014.

Minor comments:

Page 6 line 109 and 110: "low-LMW" change to low-MW, and "MW MSOM" do you mean WSOM here?

[Response]: Thanks for the reviewer's comment. "low-LMW" has been changed to "low-MW", and "MW MSOM" has been changed to "MW WSOM".

Page 10 line 193: "with a scanning speed of 12,000 nm/min".

[Response]: Yes, that is right. The scanning speed for EEM spectra collection was indeed set at 12,000 nm/min.

Page 13 line 252: "lager" did the author mean larger?

[Response]: Thanks. It has been revised.

Page 18 line 343: "were divided five fluorescence regions" to "were divided in five fluorescence regions"

[Response]: Thanks. The relevant statement has been modified.

Page 13 Figure 1: Would the small percent of LMW in HMW samples and HMW in the LMW solution influence the subsequent observation?

[Response]: Thanks for the reviewer's comment. As we replied above, the relevant influence is negligible.

Page 20 line 396: "HMW HULIS generally exhibit more intense at 1721" rephrase

[Response]: Thanks. The phrase "more intense" has been replaced with "a stronger

band".

[Revise]: Text S2 in the revised supporting information:

"HMW HULIS generally exhibit a stronger band at 1721 $cm^{-1}$ compared to LMW HULIS in both seasonal aerosols (Fig. S6)."

Page 23 line 436 : "an strong" please correct.

[Response]: Thanks. This sentence has been modified.

[Revise]: Text S2 in the revised supporting information:

"This provides a potential explanation for the higher $MAE_{365}$ values observed in winter HULI compared to summer HULIS."

Page 33 line 613: "HMW HULIS contain amounts of carboxylic functional groups, reduced nitrogen species (e.g., amines) and aromatic species than LMW HULIS", contain more/higher?

[Response]: Thanks for the suggestion. We have incorporated the term "higher" after "contain".

Supplement:

Figure S4: Use different type of markers between the summer and winter values of AAE vs MAE for the current study.

[Response]: After a thorough examination of the provided Figure S4 and associated descriptions, we find that these statements lack precision. Consequently, we have made the decision to delete this figure and the corresponding statements.

---

## Author Comment (AC2)

**Response to Editors and Reviewers**

Thanks for reviewers' careful reading and constructive comments and suggestions. We made every effort to respond to reviewers' questions point to point, and carefully revised our manuscript and Supporting Information based on their comments. To enhance clarity, we present the reviewers' comments in regular black font, while our responses are displayed in blue normal font. The modified content in both the manuscript and the Supporting Information is highlighted in red font.

**Anonymous Referee #1**

The manuscript by Fan et al. isolated low- and high- molecular weight HULIS from ambient samples collected at Anhui, China. The HULIS samples were measured for molecular size distribution, UV-VIS light absorption, fluorescence spectra, infrared absorption, and high-resolution mass spectra. Technical approaches of the sampling and chemical analysis procedures sound. Discussion of the data is occasionally speculative. The reviewer has a concern about the interpretation of the mass spectra, as detailed below.

The data shown in the manuscript would add some information to the existing literature of HULIS studies. The manuscript looks like a collection of data, rather than discussing novel findings in detail. The reviewer thinks that this manuscript may be more suitable to be considered as a measurement report, though the final decision should be made by the editor and authors.

[Response]: Thanks for the reviewer's suggestion. We have added some discussion on HULIS in the revised manuscript.

[Revise]: Lines 257-262, 326-340, 389-392, 436-444 in the revised manuscript:

"Many previous studies have demonstrated that fresh BB HULIS generally consist of small molecular-sized chromophores (Di Lorenzo et al., 2018; Di Lorenzo et al., 2017; Wong et al., 2017; Wong et al., 2019). However, a notable enhancement in the

formation of large molecular-sized chromophores has been found when they undergo intricate atmospheric processes (Di Lorenzo et al., 2018; Di Lorenzo et al., 2017; Wong et al., 2017; Wong et al., 2019)."

"These findings are consistent with the observation in our previous study that larger WSOM generally own higher MAE365 but smaller AAE values than smaller WSOM (Fan et al., 2021). Generally, the results suggest that HMW HULIS exhibit stronger light-absorbing ability but with light absorption showing a weaker wavelength dependence. It is worth noting that combustion sources, such as BB and coal combustion, usually emit primary HULIS with high MAE365 values due to the enrichment of poly-aromatic and unsaturated species (Cao et al., 2021; Fan et al., 2018; Huo et al., 2021; Zhang et al., 2021), but with small molecular weight distributions (Di Lorenzo et al., 2018; Di Lorenzo et al., 2017; Wong et al., 2017). Furthermore, subsequent pronounced photooxidation and photobleaching processes can lead to a enrichment and/or formation of large sized chromophores (Di Lorenzo et al., 2018; Di Lorenzo et al., 2017; Wong et al., 2017), concurrently resulting in a reduction in their absorption capacity and an enhancement of their spectra dependence on wavelength (Chen et al., 2021b; Fan et al., 2019; Wu et al., 2018; Wu et al., 2020; Zhang et al., 2022a)."

"It is worth noting that the ESI- HRMS could reveal molecular composition of a subset of organic molecules that are biased ionized in the negative ESI source rather than representing the entire HULIS composition (He et al., 2023; Lin et al., 2012)."

"It is noted that these values are considerably smaller than those revealed by HPSEC analysis. This difference could be explained by several factors: (1) ESI- HRMS is based towards relatively small molecules that easily protonated in negative ESI mode (He et al., 2023; Lin et al., 2012); (2) SEC can provide an apparent rather than accurate molecular size of molecules due to the lack of appropriate standards for column calibration (Fan et al., 2023; Wong et al., 2017), and (3) the potential disassembly of larger molecules stabilized by weak forces during electrospray ionization of HRMS (Fan et al., 2021; Phillips et al., 2017). Nevertheless, both HRMS and HPSEC indicate that summer HULISs exhibit larger sizes than winter HULISs."

Major comment

The size-exclusion chromatograms demonstrated that the dominant portion of HULIS have molecular weights of larger than 1000 Da. On the other hand, the mass spectrometer (orbitrap, ESI negative mode) measured the mass ranges of m/z = 60 ~ 900. The authors showed the mass spectra for the range of m/z 100 ~ (approximately) 450. The reviewer is confused how to interpret the data. The reviewer only has the knowledge on analyzing relatively small molecules using the ESI mass spectrometers. The reviewer's understanding is that fragmentation of molecules is minimal in the ESI negative mode. If the idea is also applied to the current dataset, the mass spectra measured by the study would only cover a relatively small portion of HULIS. On the other hand, if the mass spectrometer were to be measuring most of components in HULIS due to fragmentation during ionization processes, the meanings of the whole discussion in section 3.3 is unclear. The reviewer suggests the authors to clarify this point in the revised manuscript, and update the descriptions in section 3.3 if necessary.

[Response]: Thanks for the reviewer's comment. The results obtained from size-exclusion chromatograms (SEC) indeed suggest that a substantial portion of HULIS have molecular weights larger than 1000 Da, while the ESI- high-resolution mass spectra (ESI-HRMS) revealed the major molecules within HULIS in the mass range of m/z = 100~450. These results are not contradictory, primarily due to the distinct detection principles between SEC and HRMS, as well as their individual limitations.

For SEC, samples are primarily separated based on their molecular size as they pass through a stationary phase consisting of materials with varying pore sizes (Fan et al., 2023). Only molecules with a critical diameter smaller than the opening of the gel pores are allowed to enter (or be retained). Consequently, smaller solute molecules exhibit longer retention times and larger elution volumes, while larger solute molecules have shorter retention times and smaller elution volumes. Furthermore, the size of these molecules was calculated by the relationship between the retention time and the molecular weight of standard polymer molecules in solution (Wong et al.,

2017). However, due to the lack of proper standard compounds, the calculated molecular size of target samples is apparent rather than absolute (Fan et al., 2023; Wong et al., 2017).

In comparison to SEC separation processes, HRMS rarely provide signal for molecular weights above 900 Da because it is biased towards lower masses (He et al., 2023; Lin et al., 2012). In addition, HRMS with negative ESI mode is more biased towards molecules that are easily ionized by proton loss, including carboxyl, hydroxyl, carbonyl and other oxygen-containing group substances (He et al., 2023; Lin et al., 2012). Moreover, the high temperature of ESI ionization source can disrupt some intermolecular forces, leading to the detection of decomposition products of large molecules assembled with small molecules through such intermolecular forces (Fan et al., 2021; Phillips et al., 2017).

In total, the inconsistent results from SEC and HRMS might be explained by several points: (1) ESI- HRMS mainly focuses on relatively small molecules and is biased towards on those molecules easily protonated; (2) SEC can provide an apparent rather than accurate molecular size of molecules due to the lack of appropriate standards for column calibration, and (3) the potential disassembly of larger molecules stabilized by weak forces during electrospray ionization of HRMS. The distinct principles and limitations on SEC and HRMS have been added in the revised manuscript.

[Revise]: Lines 183-185, 389-392, 436-444 in the revised manuscript:

"It should be noted that the MW values estimated here are nominal rather than absolute due to the lack of appropriate standards for column calibration (Fan et al., 2023; Wong et al., 2017)."

"It is worth noting that the ESI- HRMS could reveal molecular composition of a subset of organic molecules that are biased ionized in the negative ESI source rather than representing the entire HULIS composition (He et al., 2023; Lin et al., 2012)."

"It is noted that these values are considerably smaller than those revealed by HPSEC analysis. This difference could be explained by several factors: (1) ESI- HRMS is based towards relatively small molecules that easily protonated in negative ESI mode

(He et al., 2023; Lin et al., 2012); (2) SEC can provide an apparent rather than accurate molecular size of molecules due to the lack of appropriate standards for column calibration (Fan et al., 2023; Wong et al., 2017), and (3) the potential disassembly of larger molecules stabilized by weak forces during electrospray ionization of HRMS (Fan et al., 2021; Phillips et al., 2017)."

[Reference]:

(1) Fan, X., Cai, F., Xu, C., Yu, X., Wang, Y., Xiao, X., Ji, W., Cao, T., Song, J., Peng, P.a., 2021. Molecular weight-dependent abundance, absorption, and fluorescence characteristics of water-soluble organic matter in atmospheric aerosols. Atmos. Environ. 247.

(2) Fan, X., Cheng, A., Chen, D., Cao, T., Ji, W., Song, J., Peng, P., 2023. Investigating the molecular weight distribution of atmospheric water-soluble brown carbon using high-performance size exclusion chromatography coupled with diode array and fluorescence detectors. Chemosphere 338, 139517.

(3) He, T., Wu, Y., Wang, D., Cai, J., Song, J., Yu, Z., Zeng, X., Peng, P.a., 2023. Molecular compositions and optical properties of water-soluble brown carbon during the autumn and winter in Guangzhou, China. Atmos. Environ. 296, 119573.

(4) Lin, P., Rincon, A.G., Kalberer, M., Yu, J.Z., 2012. Elemental composition of HULIS in the Pearl River Delta Region, China: results inferred from positive and negative electrospray high resolution mass spectrometric data. Environ. Sci. Technol. 46, 7454-7462.

(5) Phillips, S.M., Bellcross, A.D., Smith, G.D., 2017. Light Absorption by Brown Carbon in the Southeastern United States is pH-dependent. Environ. Sci. Technol. 51, 6782-6790.

Minor comments

Abstract: 'This observation suggests the possibility of small molecules assembling to form the HMW HULIS through intermolecular weak forces.' To the best of my

knowledge, small molecules that are assembled by intermolecular forces are not called as high-molecular weight species. They are simply clusters or aggregates of molecules. If the authors interpretation were to be right, the reviewer is not sure if HMW HULIS should really be called as 'HMW.' This point is related with the above-mentioned major comment. The reviewer suggests the authors to clarify the point.

[Response]: Thanks for the reviewer's comment. We concur with the fact that the assembly of small molecules through intermolecular forces can result in the formation of clusters or aggregates. However, it is crucial to clarify that the term of "HMW HULIS" utilized in this study represents an operational definition established through a combination of ultrafiltration (UF) and solid-phase extraction (SPE), rather than being solely based on its molecular morphology.

The UF-SPE isolation procedure for HMW HULIS is thoroughly detailed in our manuscript. In summary, we employed a stirred UF cell with a nominal-cutoff membrane of 1000 Da to isolate the water-soluble organic matters (WSOM). Consequently, the retentate solution was regarded as high-molecular-weight (HMW) WSOM, while the permeate solution was identified as low-molecular-weight (LMW) WSOM (Fan et al., 2021). Following this, both HMW and LMW WSOM fractions underwent SPE protocols, a-well established method for purifying aerosol HULIS (Fan et al., 2012; Zou et al., 2020), resulting in the isolation of HMW and LMW HULIS, respectively. It is noteworthy that the term "HMW" has been consistently employed in our prior study on aerosol WSOM (Fan et al., 2021), and is prevalent in earlier research on dissolved organic matters in marine and river environments (Du et al., 2021; Simjouw et al., 2005; Yang et al., 2021).

We appreciate the opportunity to clarify the operational nature of our definition and highlight its alignment with established terminology in the broader scientific literature.

[Reference]:
(1) Du, H., Cao, Y., Li, Z., Li, L., Xu, H., 2021. Formation and mechanisms of

hydroxyl radicals during the oxygenation of sediments in Lake Poyang, China. Water Res. 202, 117442.

(2) Fan, X.J., Song, J.Z., Peng, P.A., 2012. Comparison of isolation and quantification methods to measure humic-like substances (HULIS) in atmospheric particles. Atmos. Environ. 60, 366-374.

(3) Fan, X., Cai, F., Xu, C., Yu, X., Wang, Y., Xiao, X., Ji, W., Cao, T., Song, J., Peng, P.a., 2021. Molecular weight-dependent abundance, absorption, and fluorescence characteristics of water-soluble organic matter in atmospheric aerosols. Atmos. Environ. 247.

(4) Simjouw, J.-P., Minor, E.C., Mopper, K., 2005. Isolation and characterization of estuarine dissolved organic matter: Comparison of ultrafiltration and C18 solid-phase extraction techniques. Mar. Chem. 96, 219-235.

(5) Yang, K., Zhang, Y., Dong, Y., Li, D., Li, W., 2021. Metal binding by dissolved organic matter in hypersaline water: A size fractionation study using different isolation methods. Limnologica 87, 125849.

(6) Zou, C., Li, M., Cao, T., Zhu, M., Fan, X., Peng, S., Song, J., Jiang, B., Jia, W., Yu, C., Song, H., Yu, Z., Li, J., Zhang, G., Peng, P.a., 2020. Comparison of solid phase extraction methods for the measurement of humic-like substances (HULIS) in atmospheric particles. Atmos. Environ. 225, 117370.

L87 Previous studies have shown that aerosol WSOM, also known as brown carbon… WSOM and BrC are obviously different concepts. If WSOM and BrC are really considered as equal, please cite references to support the statement.

[Response]: Thanks for the reviewer's comment. It's acknowledged that WSOM and BrC are distinct concepts. The WSOM encompass various light-absorbing organic fractions. For the sake of studying complex BrC fractions, WSOM is often regarded as a surrogate for water-soluble brown carbon in comprehensive analyses. Consequently, the relevant statement has been revised in the manuscript.

[Revise]: Line 89 in the revised manuscript:

"Previous studies have shown that aerosol WSOM, often seen as water-soluble brown

carbon (BrC), are comprised of a continuum of molecular weight (MW) species, as revealed by high-performance exclusion chromatography (HPSEC) analysis"

Figure S1: It would be better to show the location of the observation site in the map.

[Response]: Thanks for the reviewer's suggestion. The sampling site has been labeled in Figure S1. The modified Figure S1 was shown as follow.

[Figure]

L258 'Therefore, the higher proportions of large-size chromophores and resulting larger apparent molecular size of HMW HULIS may indicate their possible secondary formation nature.' It is not clear how this statement could be supported by discussion in other parts of the manuscript. The statement seems to be a hypothesis, rather than what the authors can convincingly tell from their results.

[Response]: Thanks for the reviewer's comment. The relevant statements have been

modified in the revised manuscript.

[Revise]: Lines 257-265 in the revised manuscript:

"Many previous studies have demonstrated that fresh BB HULIS generally consist of small molecular-sized chromophores (Di Lorenzo et al., 2018; Di Lorenzo et al., 2017; Wong et al., 2017; Wong et al., 2019). However, a notable enhancement in the formation of large molecular-sized chromophores has been found when they undergo intricate atmospheric processes (Di Lorenzo et al., 2018; Di Lorenzo et al., 2017; Wong et al., 2017; Wong et al., 2019). Based on these limited studies, it is suggested that HMW HULIS, characterized by higher proportions of large-size chromophores and the resulting larger apparent molecular size, might be associated with the products from atmospheric aging process rather than being emitted directly by primary sources."

L330 'Therefore, it can be speculated that LMW HULIS are more susceptible to enrich the by-products resulting from the degradation and oxidation of BrC during processes like photooxidation and photobleaching.' Are there any other possibilities in differences (e.g., emission sources)? The reviewer suggests the authors to discuss potential sources and atmospheric processes in more detail.

[Response]: Thanks for the reviewer's comment. As indicated by previous studies, combustion sources, such as biomass burning and coal combustion, can generate amounts of primary HULIS with high $MAE_{365}$ but with small molecular weight distributions (Di Lorenzo et al., 2018; Fan et al., 2018; Wong et al., 2017). Further complex atmospheric aging processes (e.g., photooxidation and photobleaching) lead to a enrichment and/or formation of large sized chromophores (Di Lorenzo et al., 2017; Wong et al., 2017), concurrently resulting in a reduction in $MAE_{365}$ and a increase of AAE for these primary HULIS (Chen et al., 2021; Fan et al., 2019). From this perspective, the secondary organic aerosol formation might induce the generation of HMW HULIS with lower light absorption capacity and weaker light absorption wavelength dependence. In contrast, LMW HULIS is more likely to represent small-sized primary HULIS and/or by-products from degradation and oxidation of

primary large-sized HULIS. Considering the complex sources of ambient HULIS, future studies should explore the MW-dependent light absorption characteristics of HULIS from different sources.

[Revise]: Lines 330-346 in the revised manuscript:

"It is worth noting that combustion sources, such as BB and coal combustion, usually emit primary HULIS with high MAE365 values due to the enrichment of poly-aromatic and unsaturated species (Cao et al., 2021; Fan et al., 2018; Huo et al., 2021; Zhang et al., 2021), but with small molecular weight distributions (Di Lorenzo et al., 2018; Di Lorenzo et al., 2017; Wong et al., 2017). Furthermore, subsequent pronounced photooxidation and photobleaching processes can lead to a enrichment and/or formation of large sized chromophores (Di Lorenzo et al., 2018; Di Lorenzo et al., 2017; Wong et al., 2017), concurrently resulting in a reduction in their absorption capacity and an enhancement of their spectra dependence on wavelength (Chen et al., 2021b; Fan et al., 2019; Wu et al., 2018; Wu et al., 2020; Zhang et al., 2022a). From this perspective, the SOA formation might induce the generation of HMW HULIS with lower light absorption capacity and weaker light absorption wavelength dependence. In contrast, LMW HULIS is more likely to represent small-sized primary HULIS and/or by-products resulting from the degradation and oxidation of primary large-sized HULIS. Considering the complex sources of ambient HULIS, future studies should explore the MW-dependent light absorption characteristics of HULIS from different sources."

[Reference]

(1) Chen, Q., Hua, X., Dyussenova, A., 2021. Evolution of the chromophore aerosols and its driving factors in summertime Xi'an, Northwest China. Chemosphere 281, 130838.

(2) Di Lorenzo, R.A., Place, B.K., VandenBoer, T.C., Young, C.J., 2018. Composition of Size-Resolved Aged Boreal Fire Aerosols: Brown Carbon, Biomass Burning Tracers, and Reduced Nitrogen. ACS Earth and Space

Chemistry 2, 278-285.

(3) Di Lorenzo, R.A., Washenfelder, R.A., Attwood, A.R., Guo, H., Xu, L., Ng, N.L., Weber, R.J., Baumann, K., Edgerton, E., Young, C.J., 2017. Molecular-Size-Separated Brown Carbon Absorption for Biomass-Burning Aerosol at Multiple Field Sites. Environ. Sci. Technol. 51, 3128-3137.

(4) Fan, X., Li, M., Cao, T., Cheng, C., Li, F., Xie, Y., Wei, S., Song, J., Peng, P.a., 2018. Optical properties and oxidative potential of water- and alkaline-soluble brown carbon in smoke particles emitted from laboratory simulated biomass burning. Atmos. Environ. 194, 48-57.

(5) Fan, X., Yu, X., Wang, Y., Xiao, X., Li, F., Xie, Y., Wei, S., Song, J., Peng, P.a., 2019. The aging behaviors of chromophoric biomass burning brown carbon during dark aqueous hydroxyl radical oxidation processes in laboratory studies. Atmos. Environ. 205, 9-18.

(6) Wong, J.P.S., Nenes, A., Weber, R.J., 2017. Changes in Light Absorptivity of Molecular Weight Separated Brown Carbon Due to Photolytic Aging. Environ. Sci. Technol. 51, 8414-8421.

L338 'humic-like fluorophores'. I am confused about this statement. All the fluorophores in humic-like substances could be called as humic-like fluorophores. [Response]: Thanks for the reviewer's suggestion. Humic-like substances (HULIS) are atmospheric compounds resembling humic and fulvic acids found in terrestrial and aquatic systems (Graber and Rudich, 2006; Win et al., 2018). They share many functional groups, such as poly-carboxylates, carbonyls, phenols, quinones, aliphatics, and aromatics (Kumar et al., 2017; Zou et al., 2020). In practice, HULIS are typically isolated from water-soluble organic compounds (WSOM) in ambient aerosols through solid-phase extraction, making them more of an operational definition (Fan et al., 2012; Zou et al., 2020). Nevertheless, HULIS comprise complex organic compounds rather than pure humic matters.

Excitation-emission matrix (EEM) spectroscopy can provide structural information about chromophores and has been widely applied to identify the sources and chemical

nature of WSOM, including HULIS (Cao et al., 2023; Chen et al., 2016; ). It is important to note that fluorescence is a radiative process occurring between two energy levels of the same multiplicity (Andrade-Eiroa et al., 2013). Generally, compounds with rigid planar structures and highly conjugated systems exhibit intrinsic fluorescence emission characteristics. Consequently, the complex composition in HULIS results in different fluorescence region (Ex/Em) in EEM spectra. The chromophores in fluorescence spectra can be considered a "fingerprinting" tool, with distinct regions representing different components. According to many previous studies (Chen et al., 2003; Qin et al., 2018), six fluoresence regions can be identified and assigned, as detailed in the following table:

|     | Ex (nm) | Em (nm) | Assignment |
| --- | --- | --- | --- |
| I   | 200–250 | 290–330 | Aromatic proteins I |
| II  | 200–250 | 330–380 | Aromatic proteins II |
| III | 200–250 | 380–520 | Fulvic acid-like substances |
| IV  | 250–400 | 290–380 | Soluble microbial byproducts |
| V   | 250–400 | 380–520 | Humic acid-like substances |

In general, pronounced fluorescence peaks at Ex/Em = 210-235/395-410 nm could indicate a predominance of humic-like fluorophores, consistent with findings in many previous studies (Cao et al., 2023; Chen et al., 2016; Qin et al., 2018).

[Reference]

(1) Andrade-Eiroa, Á., Canle, M., Cerdá, V., 2013. Environmental Applications of Excitation-Emission Spectrofluorimetry: An In-Depth Review I. Applied Spectroscopy Reviews 48, 1-49.

(2) Cao, T., Li, M., Xu, C., Song, J., Fan, X., Li, J., Jia, W., Peng, P., 2023. Technical note: Chemical composition and source identification of fluorescent components in atmospheric water-soluble brown carbon by excitation – emission matrix spectroscopy with parallel factor analysis – potential limitations and applications. Atmos. Chem. Phys. 23, 2613-2625.

(3) Chen, Q., Ikemori, F., Higo, H., Asakawa, D., Mochida, M., 2016. Chemical Structural Characteristics of HULIS and Other Fractionated Organic Matter in Urban Aerosols: Results from Mass Spectral and FT-IR Analysis. Environ. Sci. Technol. 50, 1721-1730.

(4) Chen, W., Westerhoff, P., Leenheer, J.A., Booksh, K., 2003. Fluorescence Excitation-Emission Matrix Regional Integration to Quantify Spectra for Dissolved Organic Matter. Environ. Sci. Technol. 37, 5701-5710.

(5) Fan, X.J., Song, J.Z., Peng, P.A., 2012. Comparison of isolation and quantification methods to measure humic-like substances (HULIS) in atmospheric particles. Atmos. Environ. 60, 366-374.

(6) Graber, E.R., Rudich, Y., 2006. Atmospheric HULIS: How humic-like are they? A comprehensive and critical review. Atmos. Chem. Phys. 6, 729-753.

(7) Kumar, V., Goel, A., Rajput, P., 2017. Compositional and surface characterization of HULIS by UV-Vis, FTIR, NMR and XPS: Wintertime study in Northern India. Atmos. Environ. 164, 468-475.

(8) Qin, J., Zhang, L., Zhou, X., Duan, J., Mu, S., Xiao, K., Hu, J., Tan, J., 2018. Fluorescence fingerprinting properties for exploring water-soluble organic compounds in PM 2.5 in an industrial city of northwest China. Atmos. Environ. 184, 203-211.

(9) Win, M.S., Tian, Z., Zhao, H., Xiao, K., Peng, J., Shang, Y., Wu, M., Xiu, G., Lu, S., Yonemochi, S., Wang, Q., 2018. Atmospheric HULIS and its ability to mediate the reactive oxygen species (ROS): A review. J Environ Sci (China) 71, 13-31.

(10) Zou, C., Li, M., Cao, T., Zhu, M., Fan, X., Peng, S., Song, J., Jiang, B., Jia, W., Yu, C., Song, H., Yu, Z., Li, J., Zhang, G., Peng, P.a., 2020. Comparison of solid phase extraction methods for the measurement of humic-like substances (HULIS) in atmospheric particles. Atmos. Environ. 225, 117370.

---

## Author Response (AR2)

**Response to Editor**

Thanks for editor' constructive comments and suggestions. We have carefully revised our manuscript. To enhance clarity, we present the comments in regular black font, while our responses are displayed in blue normal font. The modified content in both the manuscript is highlighted in red font.

**Editor decision:** Publish subject to minor revisions (review by editor)

Public justification (visible to the public if the article is accepted and published):

The method and the analysis presented in this manuscript are novel, and go beyond a measurement report. So I support consideration of this manuscript as a research article. However, both reviewers raised the point about representativeness of ESI-MS analysis. In the response, the authors acknowledge these biases, but I think the discussion of these biases are too superficial. I suggest a more critical investigation of each one of the points made (SEC MW characterization being qualitative; bias towards ionizable molecules; fragmentation of larger molecules) and discuss for each one how likely that is the case, and how would it bias the results. For example, there has been some investigation of using ESI for O/C and H/C ratio calculations and they compare well to AMS measurements, so perhaps less likely.

[Response]: Thanks for the editor's insightful comments. We have addressed the viewpoints that were not thoroughly discussed in the initial response.

Regarding the qualitative nature of SEC MW characterization, we have provided a detailed discussion on how the conclusion was reached and the underlying reasons. We point out that the molecular weight calculated through HPSEC is relative and not true due to differences in molecular density, structural similarity, and potential secondary intrinsic reactions between the calibration substances PEG and HULIS. This perspective, supported by Chen et al. (2016) and Ignatev and Tuhkanen (2019), positions the results as qualitative or semi-quantitative.

Second, we have clarified the bias towards ionizable molecules in negative ESI sources. The ESI negative ion mode tends to detect specific compounds based on their inherent characteristics, ionizing functional groups within the molecule that are prone to losing protons, such as carboxylic groups, aldehydes, and hydroxyl groups (He et al., 2023; Song et al., 2022; Lin et al., 2018).

It is important to note that the ESI technique allows for the analysis of the overall molecular composition of HULIS. However, additional research methods are essential for investigating reaction mechanisms and formation processes. We have highlighted this point before introducing the method. Additionally, we have underscored the unique advantages of ESI ion sources, such as disrupting intermolecular forces like hydrogen bonds and van der Waals forces, which is valuable for detecting macromolecular complex compounds. Nevertheless, we acknowledge the need for more extensive research in this area.

Given AMS measurements, it is more likely to involve the strong fragmentation of individual compounds, which differs from the detection principles of HRMS. Consequently, their results are not directly comparable. However, this method can offer a suitable technique to delve deeper into the differences in molecular composition between HMW and LMW HULIS.

In our ongoing work, we plan to employ different technical approaches to compare HULIS obtained under identical conditions, aiming to achieve a more comprehensive understanding. We appreciate the opportunity to address these concerns and enhance the clarity and depth of our manuscript.

[Revise]: The relative statements have been incorporated into the manuscript, specifically in lines 215-219, 238-240, 262-264, 398-404, 410-413, and 424-427.

"It's important to note that the negative ESI source is more sensitive to detecting polar acid compounds, and the reported specific chemical composition here represents a fraction that is biasedly ionized in the negative ESI source, not the entire HULIS composition (He et al., 2023; Song et al., 2022)."

"Due to the potential differences in molecular densities between the calibration standards (PEG) and BB WSOC, the reported molecular weight of HULIS calculated by HPSEC herein is only estimate."

"Note that these values are estimates rather than absolute, given the absence of the

appropriate aerosol HULIS standards (Fan et al., 2023; Wong et al., 2017)."

"It is worth noting that the ESI- HRMS could reveal molecular composition of a subset of organic molecules that are biased ionized in the negative ESI source, particularly acid compounds such as carboxylic and sulfonic acids, and may not represent the entire HULIS composition (He et al., 2023; Lin et al., 2012). Additionally, HRMS techniques are often known to be biased towards low masses below 600 Da (Lin et al., 2012; Wang et al., 2019), suggesting that the molecular sizes calculated by HRMS are likely underestimated."

"The similarity in low m/z range among HMW and LMW HULIS and other bulk HULIS may indicate their shared fundamental molecules, further suggesting the potential disassembly of larger molecules."

"Moreover, SOA-derived HULIS generally contain more polar molecules (e.g., -OH, -COOH) (Di Lorenzo et al., 2017; Fan et al., 2020; Huo et al., 2021), making them more susceptible to deprotonation and leading to the generation of a greater number formulas under negative ESI conditions."

[Reference]:

(1) Chen, M., Kim, S., Park, J.-E., Jung, H.-J., Hur, J., 2016. Structural and compositional changes of dissolved organic matter upon solid-phase extraction tracked by multiple analytical tools. Analytical and Bioanalytical Chemistry 408, 6249-6258.

(2) He, T., Wu, Y., Wang, D., Cai, J., Song, J., Yu, Z., Zeng, X., Peng, P.a., 2023.

Molecular compositions and optical properties of water-soluble brown carbon during the autumn and winter in Guangzhou, China. Atmospheric Environment 296, 119573.

(3) Ignatev, A., Tuhkanen, T., 2019. Step-by-step analysis of drinking water treatment trains using size-exclusion chromatography to fingerprint and track protein-like and humic/fulvic-like fractions of dissolved organic matter. Environmental Science: Water Research & Technology 5, 1568-1581.

(4) Lin, P., Fleming, L.T., Nizkorodov, S.A., Laskin, J., Laskin, A., 2018. Comprehensive Molecular Characterization of Atmospheric Brown Carbon by High Resolution Mass Spectrometry with Electrospray and Atmospheric Pressure Photoionization. Analytical Chemistry 90, 12493-12502.

(5) Song, J., Li, M., Zou, C., Cao, T., Fan, X., Jiang, B., Yu, Z., Jia, W., Peng, P.a., 2022. Molecular Characterization of Nitrogen-Containing Compounds in Humic-like Substances Emitted from Biomass Burning and Coal Combustion. Environ. Sci. Technol. 56, 119-130.